# ONLINE HYPERPARAMETER ADAPTATION VIA AMORTIZED PROXIMAL OPTIMIZATION

## ABSTRACT

Effective performance of neural networks depends critically on effective tuning of optimization hyperparameters, especially learning rates (and schedules thereof). We present Amortized Proximal Optimization (APO), which takes the perspective that each optimization step should approximately minimize a proximal objective (similar to the ones used to motivate natural gradient and trust region policy optimization). Optimization hyperparameters are adapted to best minimize the proximal objective after one weight update. We show that an idealized version of APO (where an oracle minimizes the proximal objective exactly) achieves global convergence to stationary point and locally second-order convergence to global optimum for neural networks. APO incurs minimal computational overhead. We experiment with using APO to adapt a variety of optimization hyperparameters online during training, including (possibly layer-specific) learning rates, damping coefficients, and gradient variance exponents. For a variety of network architectures and optimization algorithms (including SGD, RMSprop, and K-FAC), we show that with minimal tuning, APO performs competitively with carefully tuned optimizers.

## 1 INTRODUCTION

Tuning optimization hyperparameters can be crucial for effective performance of a deep learning system. Most famously, carefully selected learning rate schedules have been instrumental in achieving state-of-the-art performance on challenging datasets such as ImageNet (Goyal et al., 2017) and WMT (Vaswani et al., 2017). Even algorithms such as RMSprop (Tieleman & Hinton, 2012) and Adam (Kingma & Ba, 2015), which are often interpreted in terms of coordinatewise adaptive learning rates, still have a global learning rate parameter which is important to tune. A wide variety of learning rate schedules have been proposed (Schraudolph, 1999; Li & Malik, 2016; Baydin et al., 2017). Seemingly unrelated phenomena have been explained in terms of effective learning rate schedules (van Laarhoven, 2017). Besides learning rates, other hyperparameters have been identified as important, such as the momentum decay factor (Sutskever et al., 2013), the batch size (Smith et al., 2017), and the damping coefficient in second-order methods (Martens & Grosse, 2015; Martens, 2010).

There have been many attempts to adapt optimization hyperparameters to minimize the training error after a small number of updates (Schraudolph, 1999; Andrychowicz et al., 2016; Baydin et al., 2017). This approach faces two fundamental obstacles: first, learning rates and batch sizes have been shown to affect generalization performance because stochastic updates have a regularizing effect (Dinh et al., 2017; Li et al., 2017; Mandt et al., 2017; Smith & Le, 2018; van Laarhoven, 2017). Second, minimizing the short-horizon expected loss encourages taking very small steps to reduce fluctuations at the expense of long-term progress (Wu et al., 2018). While these effects are specific to learning rates, they present fundamental obstacles to tuning *any* optimization hyperparameter, since basically any optimization hyperparameter somehow influences the size of the updates.

In this paper, we take the perspective that the optimizer's job in each iteration is to approximately minimize a proximal objective which trades off the loss on the current batch with the average change in the predictions. Specifically, we consider proximal objectives of the form $J(\phi) = h(f(g(\theta, \phi))) + \lambda \mathcal{D}(f(\theta), f(g(\theta, \phi)))$, where $f$ is a model with parameters $\theta$, $h$ is an approximation to the objective function, $g$ is the base optimizer update with hyperparameters $\phi$, and $\mathcal{D}$

is a distance metric. Indeed, approximately solving such a proximal objective motivated the natural gradient algorithm (Amari, 1998), as well as proximal reinforcement learning algorithms (Schulman et al., 2017; 2015). We introduce Amortized Proximal Optimization (APO), an approach which adapts optimization hyperparameters to minimize the proximal objective in each iteration. We use APO to tune hyperparameters of SGD, RMSprop, and K-FAC; the hyperparameters we consider include (possibly layer-specific) learning rates, damping coefficients, and the power applied to the gradient covariances.

Notice that APO has a hyperparameter $\lambda$ which controls the aggressiveness of the updates. We believe such a hyperparameter is necessary until the aforementioned issues surrounding stochastic regularization and short-horizon bias are better understood. However, in practice we find that by performing a simple grid search over $\lambda$, we can obtain automatically-tuned learning rate schedules that are competitive with manual learning rate decay schedules. Furthermore, APO can automatically adapt several optimization hyperparameters with only a single hand-tuned hyperparameter.

We provide theoretical justification for APO by proving strong convergence results for an oracle which solves the proximal objective exactly in each iteration. In particular, we show global linear convergence and locally quadratic convergence under mild assumptions. These results motivate the proximal objective as a useful target for meta-optimization.

We evaluate APO on real-world tasks including image classification on MNIST, CIFAR-10, CIFAR-100, and SVHN. We show that adapting learning rates online via APO yields faster training convergence than the best fixed learning rates for each task, and is competitive with manual learning rate decay schedules. Although we focus on fast optimization of the training objective, we also find that the solutions found by APO generalize at least as well as those found by fixed hyperparameters or fixed schedules.

## 2 AMORTIZED PROXIMAL OPTIMIZATION

We view a neural network as a parameterized function $\mathbf{z} = f(\mathbf{x}, \theta)$, where $\mathbf{x}$ is the input, $\theta$ are the weights and biases of the network, and $\mathbf{z}$ can be interpreted as the output of a regression model or the un-normalized log-probabilities of a classification model. Let the training dataset be $\{(\mathbf{x}_i, t_i)\}_{i=1}^N$, where input $\mathbf{x}_i$ is associated with target $t_i$. Our goal is to minimize the loss function:

$$\mathcal{L}(\mathbf{Z}, \mathbf{T}) = \sum_{i=1}^N \ell(\mathbf{z}_i, t_i) = \sum_{i=1}^N \ell(f(\mathbf{x}_i, \theta), t_i), \tag{1}$$

where $\mathbf{Z}$ is the matrix of network outputs on all training examples $\mathbf{x}_1, \ldots, \mathbf{x}_N$, and $\mathbf{T}$ is the vector of labels. We design an iterative optimization algorithm to minimize Eq. 1 under the following framework: in the $k$th iteration, one aims to update $\theta$ to minimize the following proximal objective:

$$h_{prox}(\theta) = h(f(\mathbf{x}, \theta)) + \lambda \, \mathbb{E}_{\tilde{\mathbf{x}} \sim \mathcal{P}}[\mathcal{D}(f(\tilde{\mathbf{x}}, \theta), f(\tilde{\mathbf{x}}, \theta_k))], \tag{2}$$

where $\mathbf{x}$ is the data used in the current iteration, $\mathcal{P}$ is the distribution of data, $\theta_k$ is the parameters of the neural network at the current iteration, $h(\cdot)$ is some approximation of the loss function, and $\mathcal{D}(\cdot, \cdot)$ represents the distance between network outputs under some metric (for notational convenience, we use mini-batch size of 1 to describe the algorithm). We first provide the motivation for this proximal objective in Section 2.1; then in Section 2.2, we propose an algorithm to optimize it in an online manner.

### 2.1 MOTIVATION FOR THE PROXIMAL OBJECTIVE

In this section, we show that by approximately minimizing simple instances of Eq. 2 in each iteration (similar to Schulman et al. (2015)), one can recover the classic Gauss-Newton algorithm and Natural Gradient Descent (Amari, 1998). In general, updating $\theta$ so as to minimize the proximal objective is impractical due to the complicated nonlinear relationship between $\theta$ and $\mathbf{z}$. However, one can find an approximate solution by linearizing the network function:

$$f(\mathbf{x}, \theta + \Delta\theta) \approx f(\mathbf{x}, \theta) + \mathbf{J}\Delta\theta, \tag{3}$$

where $\mathbf{J} = \nabla_\theta f(\mathbf{x}, \theta)$ is the Jacobian matrix. We consider the following instance of Eq. 2:

$$h_{prox}(\theta) = \Delta\mathbf{z}^\top \nabla_\mathbf{z} \ell(f(\mathbf{x}, \theta_k), t) + \lambda \, \mathbb{E}_{\tilde{\mathbf{x}} \sim \mathcal{P}}[\mathcal{D}(f(\tilde{\mathbf{x}}, \theta), f(\tilde{\mathbf{x}}, \theta_k))], \tag{4}$$

where $\Delta \mathbf{z} \triangleq f(\mathbf{x}, \theta) - f(\mathbf{x}, \theta_k)$ is the change of network output, $t$ is the label of current data $\mathbf{x}$. Here $h(\cdot)$ is defined as the first-order Taylor approximation of the loss function. Using the linear approximation (Eq. 3), and a local second-order approximation of $\mathcal{D}$, this proximal objective can be written as:

$$h_{prox}(\theta) \approx \Delta \theta^\top \nabla_\theta \ell(f(\mathbf{x}, \theta_k), t) + \lambda \Delta \theta^\top \mathbb{E}_{\tilde{\mathbf{x}} \sim \mathcal{P}} \left[ \tilde{\mathbf{J}}^\top \nabla^2 \tilde{\mathcal{D}} \tilde{\mathbf{J}} \right] \Delta \theta, \tag{5}$$

where $\tilde{\mathbf{J}} = \nabla_\theta f(\tilde{\mathbf{x}}, \theta_k)$ is the Jacobian matrix on data $\tilde{\mathbf{x}}$, $\nabla^2 \tilde{\mathcal{D}} \triangleq \nabla_{\tilde{\mathbf{z}}}^2 \mathcal{D}(\tilde{\mathbf{z}}, f(\tilde{\mathbf{x}}, \theta_k))$ is the Hessian matrix of the dissimilarity measured at $\tilde{\mathbf{z}} = f(\tilde{\mathbf{x}}, \theta_k)$.

Solving Eq. 5 yields:

$$\Delta \theta = -\frac{1}{\lambda} \mathbf{G}^{-1} \nabla_\theta \ell(f(\mathbf{x}, \theta), t), \tag{6}$$

where $\mathbf{G} \triangleq \mathbb{E}_{\tilde{\mathbf{x}} \sim \mathcal{P}} \left[ \tilde{\mathbf{J}}^\top \nabla^2 \tilde{\mathcal{D}} \tilde{\mathbf{J}} \right]$ is the pre-conditioning matrix. Different settings for the dissimilarity term $\mathcal{D}$ yield different algorithms. When

$$\mathcal{D}(\tilde{\mathbf{z}}, \tilde{\mathbf{z}}_k) = \|\tilde{\mathbf{z}} - \tilde{\mathbf{z}}_k\|_2^2 \tag{7}$$

is defined as the squared Euclidean distance, Eq. 6 recovers the classic Gauss-Newton algorithm. When

$$\mathcal{D}(\tilde{\mathbf{z}}, \tilde{\mathbf{z}}_k) = \ell(\tilde{\mathbf{z}}) - \ell(\tilde{\mathbf{z}}_k) - \langle \nabla \ell(\tilde{\mathbf{z}}_k), \tilde{\mathbf{z}} - \tilde{\mathbf{z}}_k \rangle \tag{8}$$

is defined as the Bregman divergence, Eq. 6 yields the Generalized Gauss-Newton (GGN) method. When the output of neural network parameterizes an exponential-family distribution, the dissimilarity term can be defined as Kullback-Leibler divergence:

$$\mathcal{D}(\tilde{\mathbf{z}}, \tilde{\mathbf{z}}_k) = \sum_y p(y|\tilde{\mathbf{z}}) \log \frac{p(y|\tilde{\mathbf{z}})}{p(y|\tilde{\mathbf{z}}_k)}, \tag{9}$$

in which case Eq. 6 yields Natural Gradient Descent (Amari, 1998). Since different versions of our proximal objective lead to various efficient optimization algorithms, we believe it is a useful target for meta-optimization.

## 2.2 Amortized optimization

Although optimizers including the Gauss-Newton algorithm and Natural Gradient Descent can be seen as ways to approximately solve Eq. 2, they rely on a local linearization of the neural network and usually require more memory and more careful tuning in practice. We propose to instead directly minimize Eq. 2 in an online manner.

Finding good hyperparameters (e.g., the learning rate for SGD) is a challenging problem in practice. We propose to adapt these hyperparameters online in order to best optimize the proximal objective. Consider any optimization algorithm (*base-optimizer*) of the following form:

$$\theta \leftarrow g(\mathbf{x}, \mathbf{t}, \theta, \xi, \phi). \tag{10}$$

Here, $\theta$ is the set of model parameters, $\mathbf{x}$ is the data used in this iteration, $\mathbf{t}$ is the corresponding label, $\xi$ is a vector of statistics computed online during optimization, and $\phi$ is a vector of optimization hyperparameters to be tuned. For example, $\xi$ contains the exponential moving averages of the squared gradients of the parameters in RMSprop. $\phi$ usually contains the learning rate (global or layer-specific), and possibly other hyperparameters dependent on the algorithm.

For each step, we formulate the meta-objective from Eq. 2 as follows (for notational convenience we omit variables other than $\theta$ and $\phi$ of $g$):

$$J(\phi) = h(f(\mathbf{x}, g(\theta, \phi))) + \lambda \mathbb{E}_{\tilde{\mathbf{x}} \sim P}[\mathcal{D}(f(\tilde{\mathbf{x}}, g(\theta, \phi)), f(\tilde{\mathbf{x}}, \theta))]. \tag{11}$$

Here, $\tilde{\mathbf{x}}$ is a random mini-batch sampled from the data distribution $\mathcal{P}$. We compute the approximation to the loss, $h$, using the same mini-batch as the gradient of the base optimizer, to avoid the short horizon bias problem (Wu et al., 2018); we measure $\mathcal{D}$ on a different mini-batch to avoid instability that would result if we took a large step in a direction that is unimportant for the current batch, but important for other batches. The hyperparameters $\phi$ are optimized using a stochastic gradient-based algorithm (the *meta-optimizer*) using the gradient $\nabla_\phi J(\phi)$ (similar in spirit to (Schraudolph, 1999; Maclaurin et al., 2015)). We refer to our framework as Amortized Proximal Optimization (APO). The simplest version of APO, which uses SGD as the meta-optimizer, is shown in Algorithm 1. One can choose any meta-optimizer; we found that RMSprop was the most stable and best-performing meta-optimizer in practice, and we used it for all our experiments.

---

**Algorithm 1:** Amortized Proximal Optimization (SGD as meta-optimizer)

---

**Input** : $\eta, \theta_0, \phi_0, M, T$
**Output:** $\theta$
$\theta \leftarrow \theta_0$
$\phi \leftarrow \phi_0$
**for** $i \leftarrow 1, \ldots, M$ **do**
    sample data $(\mathbf{x}, t)$
    **for** $t \leftarrow 1, \ldots, T$ **do**
        sample $\tilde{\mathbf{x}} \sim \mathcal{P}$
        $J(\phi) = h(f(\mathbf{x}, g(\mathbf{x}, t, \theta, \xi, \phi))) + \lambda \mathcal{D}(f(\mathbf{x}, g(\mathbf{x}, t, \theta, \xi, \phi)), f(\tilde{\mathbf{x}}, \theta))$
        $\phi \leftarrow \phi - \eta \nabla_\phi J(\phi)$
    **end**
    $\theta \leftarrow g(\mathbf{x}, t, \theta, \xi, \phi)$
**end**
return $\theta$

---

# 3 ANALYSIS OF AN IDEALIZED VERSION

When considering optimization meta-objectives, it is useful to analyze idealized versions where the meta-objective is optimized exactly (even when doing so is prohibitively expensive in practice). For instance, Wu et al. (2018) analyzed an idealized SMD algorithm, showing that even the idealized version suffered from short-horizon bias. In this section, we analyze two idealized versions of APO where an oracle is assumed to minimize the proximal objective exactly in each iteration. In both cases, we obtain strong convergence results, suggesting that our proximal objective is a useful target for meta-optimization.

We view the problem in output space (i.e., explicitly designing an update schedule for $z_i$). Consider the space of outputs on all training examples; when we train a neural network, we are optimizing over a manifold in this space:

$$\mathcal{M} = \{(f(x_1, \theta), f(x_2, \theta), \ldots, f(x_N, \theta)) \mid \theta \in \mathbb{R}^D\} \tag{12}$$

We assume that $f$ is continuous, so that $\mathcal{M}$ is a continuous manifold. Given an oracle that for each iteration exactly minimizes the expectation of proximal objective Eq. 2 over the dataset, we can write one iteration of APO in output space as:

$$\mathbf{Z} \leftarrow \underset{\mathbf{Z} \in \mathcal{M}}{\arg\min} \sum_{i=1}^{N} [h(\mathbf{z}_i) + \lambda \mathcal{D}(\mathbf{z}_i, \mathbf{z}_{k,i})], \tag{13}$$

where $\mathbf{z}_i$ is $i$th column of $\mathbf{Z}$, corresponding to the network output on data $\mathbf{x}_i$ after update, $\mathbf{z}_{k,i}$ is the current network output on data $\mathbf{x}_i$.

## 3.1 PROJECTED GRADIENT DESCENT

We first define the proximal objective as Eq. 4, using the Euclidean distance as the dissimilarity measure, which corresponds to Gauss-Newton algorithm under the linearization of network. With an oracle, this proximal objective leads to projected gradient descent:

$$\mathbf{Z}_{k+1} \leftarrow \underset{\mathbf{Z} \in \mathcal{M}}{\arg\min} \left\| \mathbf{Z} - \left( \mathbf{Z}_k - \frac{1}{2\lambda} \nabla_{\mathbf{Z}} \mathcal{L}(\mathbf{Z}_k, \mathbf{T}) \right) \right\|_F^2. \tag{14}$$

Consider a loss function on one data point $\ell(\mathbf{z}) : \mathbb{R}^d \to \mathbb{R}$, where $d$ is the dimension of neural network's output. [1] We say the gradient is $L$-Lipschitz if:

$$\|\nabla_{\mathbf{z}} \ell(\mathbf{z}_1) - \nabla_{\mathbf{z}} \ell(\mathbf{z}_2)\| \leq L \|\mathbf{z}_1 - \mathbf{z}_2\|. \tag{A1}$$

When the manifold $\mathcal{M}$ is smooth, a curve in $\mathcal{M}$ is called *geodesic* if it is the shortest curve connecting the starting and the ending point. We say $\mathcal{M}$ have a $\mathcal{C}$-*bounded curvature* if for each trajectory

---

[1] For convenience of notation, we omit the dependence of loss on the fixed label.

$v(t) : [0, 1] \to \mathbb{R}^d$ going along some geodesic and $\|\dot{v}(t)\|_2 = 1$, there is $\|\ddot{v}(t)\| \leq \mathcal{C}$ with spectral norm. For each point $\mathbf{Z} \in \mathcal{M}$, consider the tangent space at point $\mathbf{Z}$ as $\mathcal{T}_{\mathbf{Z}}\mathcal{M}$. We call the projection of $\nabla \mathcal{L}(\mathbf{Z})$ onto the hyperplane $\mathcal{T}_{\mathbf{Z}}\mathcal{M}$ as the *effective gradient* of $\mathcal{L}$ at $\mathbf{Z} \in \mathcal{M}$. It is worth noting that zero effective gradient corresponds to stationary point of the neural network.

We have the following theorem stating the global convergence of Eq. 14 to stationary point:

**Theorem 1.** *Assume the loss satisfies A1. Furthermore, assume $\mathcal{L}$ is lower bounded by $\mathcal{L}^*$ and has gradient norm upper bound $\mathcal{G}$. Let $g_T^*$ be the effective gradient in the first $T$ iterations with minimal norm. When the manifold is smooth with $\mathcal{C}$-bounded curvature, with $\lambda \geq \{\mathcal{C}\mathcal{G}, \frac{L}{4}\}$, the norm of $g_T^*$ converges with rate $O\left(\frac{1}{T}\right)$ as:*

$$\|g_T^*\|_2^2 \leq \frac{16\lambda}{T} \left[\mathcal{L}(\mathbf{Z}_0) - \mathcal{L}^*\right]. \tag{15}$$

This convergence result differs from usual neural network convergence results, because here the Lipschitz constants are defined for the output space, so they are known and generally nice. For instance, $L = 1$ when we use a quadratic loss. In contrast, the gradient is in general not Lipschitz continuous in weight space for deep networks.

## 3.2 PROXIMAL NEWTON METHOD

We further replace the dissimilarity term with:

$$\mathcal{D}(\mathbf{z}_i, \mathbf{z}_{k,i}) = (\mathbf{z}_i - \mathbf{z}_{k,i})^\top \nabla^2 \ell(\mathbf{z}_{k,i})(\mathbf{z}_i - \mathbf{z}_{k,i}), \tag{16}$$

which is the second-order approximation of Eq. 8. With a proximal oracle, this variant of APO turns out to be Proximal Newton Method in the output space, if we set $\lambda = \frac{1}{2}$:

$$\mathbf{Z}_{k+1} \leftarrow \underset{\mathbf{Z} \in \mathcal{M}}{\arg\min} \left[\langle \nabla_{\mathbf{Z}} \mathcal{L}(\mathbf{Z}_k), \mathbf{Z} - \mathbf{Z}_k \rangle + \frac{1}{2} \|\mathbf{Z} - \mathbf{Z}_k\|_H^2\right], \tag{17}$$

where $\|\mathbf{Z} - \mathbf{Z}_k\|_H^2$ is the norm with local Hessian as metric. In general, Newton's method can't be applied directly to neural nets in weight space, because it is nonconvex (Dauphin et al., 2014). However, Proximal Newton Method in output space can be efficient given a strongly convex loss function.

Consider a loss $\ell(\mathbf{z})$ with $\mu$-strongly convex:

$$\ell(\mathbf{z}) - \ell(\mathbf{z}^*) \geq \frac{\mu}{2} \|\mathbf{z} - \mathbf{z}^*\|^2, \tag{A2}$$

where $\mathbf{z}^*$ is the unique minimizer and $\mu$ is some positive real number, and $L_H$-smooth Hessian: for any vector $v \in \mathbb{R}^d$ such that $\|v\| = 1$, there is:

$$\left\|\nabla_{\mathbf{z}}^2 \ell(\mathbf{z}_1)v - \nabla_{\mathbf{z}}^2 \ell(\mathbf{z}_2)v\right\| \leq L_H \|\mathbf{z}_1 - \mathbf{z}_2\|. \tag{A3}$$

The following theorem suggests the locally fast convergence rate of iteration Eq. 17:

**Theorem 2.** *Under assumptions A2 and A3, if the unique minimum $\mathbf{Z}^* \in \mathcal{M}$, then whenever iteration (17) converges to $\mathbf{Z}^*$, it converges locally quadratically[2]:*

$$\lim_{k \to \infty} \frac{\|\mathbf{Z}_{k+1} - \mathbf{Z}^*\|}{\|\mathbf{Z}_k - \mathbf{Z}^*\|^2} \leq \frac{L_H}{\mu}.$$

Hence, the proximal oracle achieves second-order convergence for neural network training under fairly reasonable assumptions. Of course, we don't expect practical implementations of APO (or any other practical optimization method for neural nets) to achieve the second-order convergence rates, but we believe the second-order convergence result still motivates our proximal objective as a useful target for meta-optimization.

---

[2]It is worth noting that here we don't need assumptions of manifold $\mathcal{M}$ being even differentiable. Therefore, the result holds for neural networks where non-smooth activation functions like ReLU are used.

## 4 RELATED WORK

Finding good optimization hyperparameters is a longstanding problem (Bengio, 2012). Classic methods for hyperparameter optimization, such as grid search, random search, and Bayesian optimization (Snoek et al., 2012; 2015; Swersky et al., 2014), are expensive, as they require performing many complete training runs, and can only find fixed hyperparameter values (e.g., a constant learning rate). Hyperband (Li et al., 2016) can reduce the cost by terminating poorly-performing runs early, but is still limited to finding fixed hyperparameters. Population Based Training (PBT) (Jaderberg et al., 2017) trains a population of networks simultaneously, and throughout training it terminates poorly-performing networks, replaces their weights by a copy of the weights of a better-performing network, perturbs the hyperparameters, and continues training from that point. PBT can find a coarse-grained learning rate schedule, but because it relies on random search, it is far less efficient than gradient-based meta-optimization.

There have been a number of approaches to gradient-based adaptation of learning rates. Gradient-based optimization algorithms can be unrolled as computation graphs, allowing the gradients of hyperparameters such as learning rates to be computed via automatic differentiation. Maclaurin et al. (2015) propagate gradients through the full unrolled training procedure to find optimal learning rate schedules offline. Stochastic meta-descent (SMD) (Schraudolph, 1999) adapts hyperparameters online. Hypergradient descent (HD) (Baydin et al., 2017) takes the gradient of the learning rate with respect to the optimizer update in each iteration, to minimize the expected loss in the next iteration. In particular, HD suffers from short horizon bias (Wu et al., 2018), while in Appendix F we show that APO does not.

Some authors have proposed learning entire optimization algorithms (Li & Malik, 2016; 2017; Andrychowicz et al., 2016). Li & Malik (2016) view this problem from a reinforcement learning perspective, where the state consists of the objective function $\mathcal{L}$ and the sequence of prior iterates $\{\theta_t\}$ and gradients $\{\nabla_\theta \mathcal{L}(\theta_t)\}$, and the action is the step $\Delta\theta$. In this setting, the update rule $\phi$ is a policy, which can be found via policy gradient methods (Sutton et al., 2000). Approaches that learn optimizers must be trained on a *set* of objective functions $\{f_1, \dots, f_n\}$ drawn from a distribution $\mathcal{F}$; this setup can be restrictive if we only have one instance of an objective function. In addition, the initial phase of training the optimizer on a distribution of functions can be expensive. APO requires only the objective function of interest and finds learning rate schedules in a single training run.

In principle, APO could be used to learn a full optimization algorithm; however, learning such an algorithm would be just as hard as the original optimization problem, so one would not expect an out-of-the-box meta-optimizer (such as RMSprop with learning rate 0.001) to work as well as it does for adapting few hyperparameters.

## 5 EXPERIMENTS

In this section, we evaluate APO empirically on a variety of learning tasks; Table 1 gives an overview of the datasets, model architectures, and base optimizers we consider.

In our proximal objective, $J(\phi) = h(f(\mathbf{x}, g(\theta, \phi))) + \lambda \mathbb{E}_{\tilde{\mathbf{x}} \sim \mathcal{P}}[\mathcal{D}(f(\tilde{\mathbf{x}}, g(\theta, \phi)), f(\tilde{\mathbf{x}}, \theta))]$, $h$ can be any approximation to the loss function (e.g., a linearization); in our experiments, we directly used the loss value $h = \ell$, as we found this to work well in many settings. As the dissimilarity term $\mathcal{D}$, we used the squared Euclidean norm.

We used APO to tune the optimization hyperparameters of four base-optimizers: SGD, SGD with Nesterov momentum (denoted SGDm), RMSprop, and K-FAC. For SGD, the only hyperparameter is the learning rate; we consider both a single, global learning rate, as well as per-layer learning rates. For SGDm, the update rule is given by:

$$v_t \leftarrow \mu v_{t-1} + g_t \tag{18}$$
$$\theta_t \leftarrow \theta_{t-1} - \eta(g_t + \mu v_t) \tag{19}$$

where $g = \nabla \ell$. Since adapting $\mu$ requires considering long-term performance (Sutskever et al., 2013), it is not appropriate to adapt it with a one-step objective like APO. Instead, we just adapt the learning rate with APO as if there's no momentum, but then apply momentum with $\mu = 0.9$ on top of the updates.

| Dataset | Architecture | Optimizer | Batch Size |
|---|---|---|---|
| Rosenbrock & 2-Layer Linear | - | RMSprop | - |
| MNIST | MLP | SGD, RMSprop | 100 |
| FashionMNIST | 2-Layer CNN | RMSprop, K-FAC | 100 |
| CIFAR-10 | ResNet-34 | SGD, RMSprop, K-FAC | 128 |
| CIFAR-100 | ResNet-34 | SGD | 128 |
| SVHN | ResNet-18 | RMSprop | 128 |

Table 1: Summary of the datasets, model architectures, and optimizers we investigate.

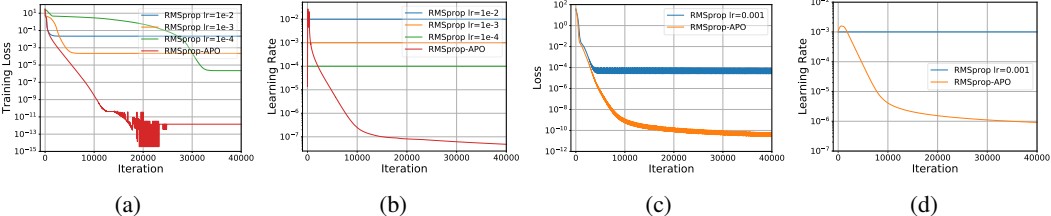

$$\begin{array}{cccc} \text{(a)} & \text{(b)} & \text{(c)} & \text{(d)} \end{array}$$

Figure 1: **Experiments on toy examples.** (a) Rosenbrock objective values during training; (b) Learning rates for RMSprop compared to the adaptive learning rate of RMSprop-APO on Rosenbrock; (c) Loss on the badly-conditioned regression problem; (d) Learning rate adaptation on the badly-conditioned regression problem.

For RMSprop, the optimizer step is given by:

$$s_t \leftarrow \gamma s_{t-1} + (1 - \gamma) g_t^2 \tag{20}$$

$$\theta_t \leftarrow \theta_{t-1} - \frac{\eta}{s_t^\rho + \epsilon} g_t \tag{21}$$

We note that, in addition to the learning rate $\eta$, we can also consider adapting $\epsilon$ and the power to which $s$ is raised in the denominator of Eq. 21—we denote this parameter $\rho$, where in standard RMSprop we have $\rho = \frac{1}{2}$. Both $\epsilon$ and $\rho$ can be interpreted as having a damping effect on the update.

K-FAC is an approximate natural gradient method (Amari, 1998) based on preconditioning the gradient by an approximation to the Fisher matrix, $\theta \leftarrow \theta - F^{-1}\nabla\ell$. For K-FAC, we tune the global learning rate and the damping factor.

**Meta-Optimization Setup.** Throughout this section, we use the following setup for meta-optimization: we use RMSprop as the meta-optimizer, with learning rate 0.1, and perform 1 meta-optimization update for every 10 steps of the base optimization. We show in Appendix E that with this default configuration, APO is robust to the initial learning rate of the base optimizer. Each meta-optimization step takes approximately the same amount of computation as a base optimization step; by performing meta-updates once per 10 base optimization steps, the computational overhead of using APO is just a small fraction more than the original training procedure.

## 5.1 TOY OPTIMIZATION PROBLEMS

**Rosenbrock.** We first validated APO on the two-dimensional Rosenbrock function, $f(x, y) = (1 - x)^2 + 100(y - x^2)^2$, with initialization $(x, y) = (1, -1.5)$. We used APO to tune the learning rate of RMSprop, and compared to standard RMSprop with several fixed learning rates. Because this problem is deterministic, we set $\lambda = 0$ for APO. Figure 1(a) shows that RMSprop-APO was able to achieve a substantially lower objective value than the baseline RMSprop. The learning rates for each method are shown in Figure 1(b); we found that APO first increases the learning rate to make rapid progress at the start of optimization, and then gradually decreases it as it approaches the local optimum. In Appendix D we show that APO converges quickly from many different locations on the Rosenbrock surface, and in Appendix E we show that APO is robust to the initial learning rate of the base optimizer.

**Badly-Conditioned Regression.** Next, we evaluated APO on a badly-conditioned regression problem (Recht & Rahimi, 2017), which is intended to be a difficult test problem for optimization algorithms. In this problem, we consider a dataset of input/output pairs $\{(x, y)\}$, where the outputs are given by $y = Ax$, where $A$ is an ill-conditioned matrix with $\kappa(A) = 10^{10}$. The task

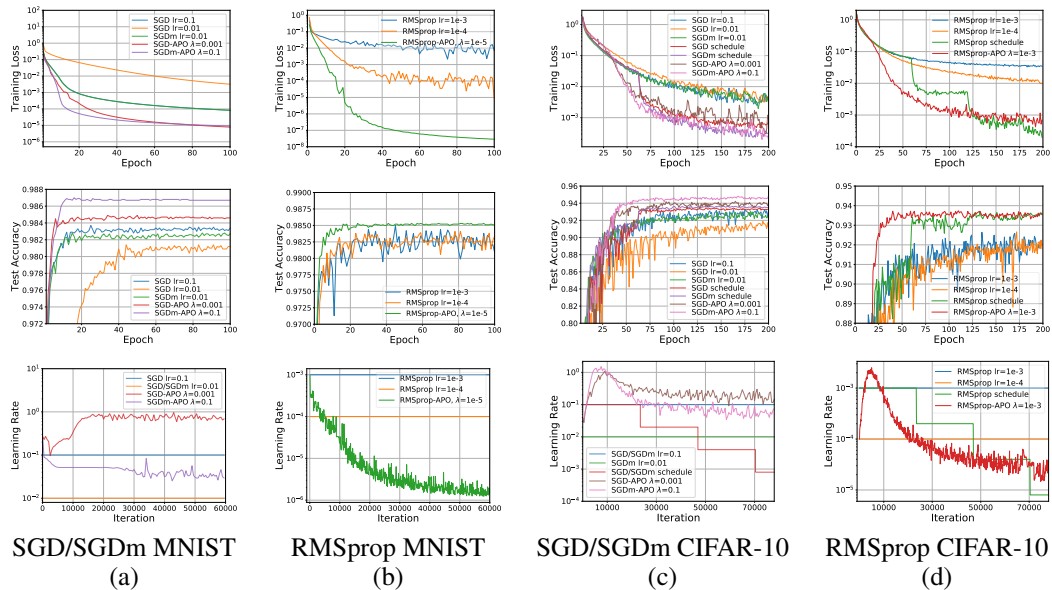

| SGD/SGDm MNIST | RMSprop MNIST | SGD/SGDm CIFAR-10 | RMSprop CIFAR-10 |
| (a) | (b) | (c) | (d) |

Figure 2: **Experiments on MNIST and CIFAR-10, with and without APO. Upper row:** mean loss over the training set. **Middle row:** accuracy on the test set. **Bottom row:** learning rate per iteration. We use ResNet34 for CIFAR-10, and train for 200 epochs with learning rate decayed by a factor of 10 every 60 epochs.

is to fit a two-layer linear model $f(x) = W_2 W_1 x$ to this data; the loss to be minimized is $\mathcal{L} = \mathbb{E}_{x \sim \mathcal{N}(0, I)} \left[ ||Ax - W_2 W_1 x||_2^2 \right]$. Figure 1(c) compares the performance of RMSprop with a hand-tuned fixed learning rate to the performance of RMSprop-APO, with learning rates shown in Figure 1(d). Again, the adaptive learning rate enabled RMSprop-APO to achieve a loss value orders of magnitude smaller than that achieved by RMSprop with a fixed learning rate.

## 5.2 REAL-WORLD DATASETS

For each of the real-world datasets we consider—MNIST, CIFAR-10, CIFAR-100, SVHN, and FashionMNIST—we chose the learning rates for the baseline optimizers via grid searches: for SGD and SGDm, we performed a grid search over learning rates $\{0.1, 0.01, 0.001\}$, while for RMSprop, we performed a grid search over learning rates $\{0.01, 0.001, 0.0001\}$. For SGD-APO and SGDm-APO, we set the initial learning rate to $0.1$, while for RMSprop-APO, we set the initial learning rate to $0.0001$. These initial learning rates are used for convenience; we show in Appendix E that APO is robust to the choice of initial learning rate. The only hyperparameter we consider for APO is the value of $\lambda$: for SGD-APO and SGDm-APO, we select the best $\lambda$ from a grid search over $\{0.1, 0.01, 1e\text{-}3\}$; for RMSprop, we choose $\lambda$ from a grid search over $\{0.1, 0.01, 1e\text{-}3, 1e\text{-}4, 1e\text{-}5, 0\}$. Note that because each value of $\lambda$ yields a *learning rate schedule*, performing a search over $\lambda$ is much more effective than searching over fixed learning rates. In particular, we show that the adaptive learning rate schedules discovered by APO are competitive with manual learning rate schedules.

### 5.2.1 MULTI-LAYER PERCEPTRON ON MNIST

First, we compare SGD and RMSprop with their APO-tuned variants on MNIST, and show that APO outperforms fixed learning rates. As the classification network for MNIST, we used a two-layer MLP with 1000 hidden units per layer and ReLU nonlinearities. We trained on mini-batches of size 100 for 100 epochs.

**SGD with APO.** We used APO to tune the global learning rate of SGD and SGD with Nesterov momentum (denoted SGDm) on MNIST, where the momentum is fixed to $0.9$. For baseline SGDm, we used learning rate $0.01$, while for baseline SGD, we used both learning rates $0.1$ and $0.01$. The training curve of SGD with learning rate $0.1$ almost coincides with that of SGDm with learning rate $0.01$. For SGD-APO, the best $\lambda$ was 1e-3, while for SGDm-APO, the best $\lambda$ was $0.1$. A comparison of the algorithms is shown in Figure 2(a). APO substantially improved the training loss for both SGD and SGDm.

Figure 3: **K-FAC results on CIFAR-10.** We compare K-FAC with a fixed learning rate and a manual learning rate schedule to APO, used to tune 1) the learning rate; and 2) both the learning rate and damping coefficient.

**RMSprop with APO.** Next, we used APO to tune the global learning rate of RMSprop. For baseline RMSprop, the best fixed learning rate was 1e-4, while for RMSprop-APO, the best $\lambda$ was 1e-5. Figure 2(b) compares RMSprop and its APO-tuned variant on MNIST. RMSprop-APO achieved a training loss about three orders of magnitude smaller than the baseline.

### 5.2.2 CIFAR-10 EXPERIMENTS

We trained a 34-layer residual network (ResNet34) (He et al., 2016) on CIFAR-10 (Krizhevsky & Hinton, 2009), using mini-batches of size 128, for 200 epochs. We used batch normalization and standard data augmentation (horizontal flipping and cropping). For each optimizer, we compare APO to 1) fixed learning rates; and 2) manual learning rate decay schedules.

**SGD with APO.** For SGD, we used both learning rates 0.1 and 0.01 since both work well. For SGD with momentum, we used learning rate 0.01. We also consider a manual schedule for both SGD and SGDm: starting from learning rate 0.1, and we decay it by a factor of 5 every 60 epochs. For the APO variants, we found that $\lambda$=1e-3 was best for SGD, while $\lambda = 0.1$ was best for SGDm. As shown in Figure 2(c), APO not only accelerates training, but also achieves higher accuracy on the test set at the end of training.

**RMSprop with APO.** For RMSprop, we use fixed learning rates 1e-3 and 1e-4, and we consider a manual learning rate schedule in which we initialize the learning rate to 1e-3 and decay by a factor of 5 every 60 epochs. For RMSprop-APO, we used $\lambda = 1e-3$. The training curves, test accuracies, and learning rates for RMSprop and RMSprop-APO on CIFAR-10 are shown in Figure 2(d). We found that APO achieved substantially lower training loss than fixed learning rates, and was competitive with the manual decay schedule. In particular, both the final training loss and final test accuracy achieved by APO are similar to those achieved by the manual schedule.

**K-FAC with APO.** We also used APO to tune the learning rate and damping coefficient of K-FAC. Similarly to the previous experiments, we use K-FAC to optimize a ResNet34 on CIFAR-10. We used mini-batches of size 128 and trained for 100 epochs. For the baseline, we used a fixed learning rate of 1e-3 as well as a decay schedule with initial learning rate 1e-3, decayed by a factor of 10 at epochs 40 and 80. For APO, we used $\lambda = 1e-2$. In experiments where the damping is not tuned, it is fixed at 1e-3. The results are shown in Figure 3. We see that K-FAC-APO performs competitively with the manual schedule when tuning just the global learning rate, and that both training loss and test accuracy improve when we tune both the learning rate and damping coefficient simultaneously.

### 5.2.3 CIFAR-100 EXPERIMENTS

Next, we evaluated APO on the CIFAR-100 dataset. Similarly to our experiments on CIFAR-10, we used a ResNet34 network with batch-normalization and data augmentation, and we trained on mini-batches of size 128, for 200 epochs. We compared SGD-APO/SGDm-APO to standard SGD/SGDm using (1) a fixed learning rate found by grid search; (2) a custom learning rate schedule in which the learning rate is decayed by a factor of 5 at epochs 60, 120, and 180. We set $\lambda = 1e-3$ for SGD-APO and $\lambda = 0.1$ for SGDm-APO. Figure 4 shows the training loss, test accuracy, and the tuned learning rate. It can be seen that APO generally achieves smaller training loss and higher test accuracy.

### 5.2.4 SVHN EXPERIMENTS

We also used APO to train an 18-layer residual network (ResNet18) with batch normalization on the SVHN dataset (Netzer et al., 2011). Here, we used the standard train and test sets, without additional training data. We used mini-batches of size 128 and trained our networks for 160 epochs. We compared APO to 1) fixed learning rates, and 2) a manual schedule in which we initialize the learning rate to 1e-3 and decay by a factor of 10 at epochs 80 and 120. We show the training loss, test accuracy, and learning rates for each method in Figure 5. Here, RMSprop-APO achieves similar

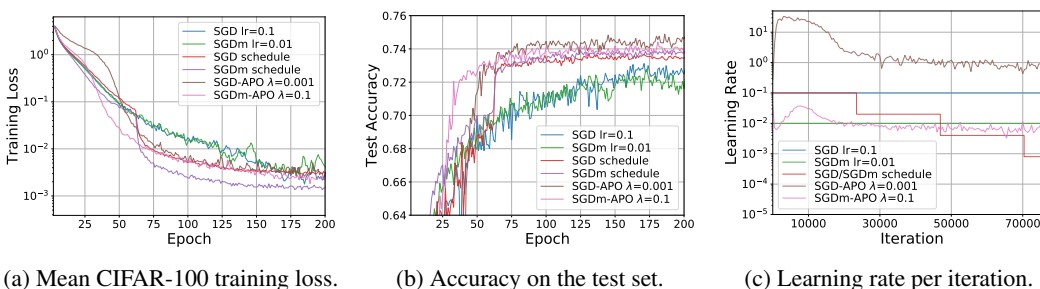

(a) Mean CIFAR-100 training loss.  (b) Accuracy on the test set.  (c) Learning rate per iteration.

Figure 4: **Comparison of SGD and SGD-APO on CIFAR-100.**

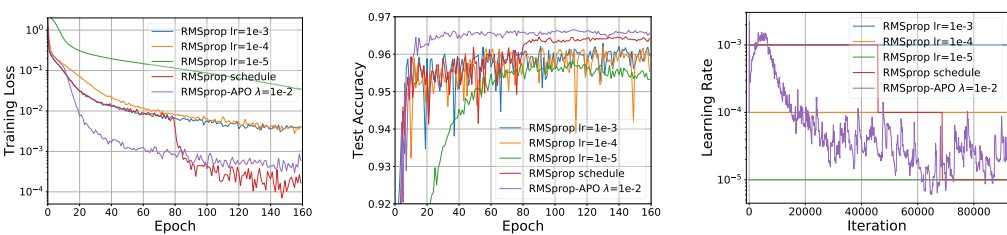

Figure 5: **Comparison of RMSprop and RMSprop-APO used to optimize a ResNet on SVHN.**

training loss to the manual schedule, and obtains higher test accuracy than the schedule. We also see that the learning rate adapted by APO spans two orders of magnitude, similar to the manual schedule.

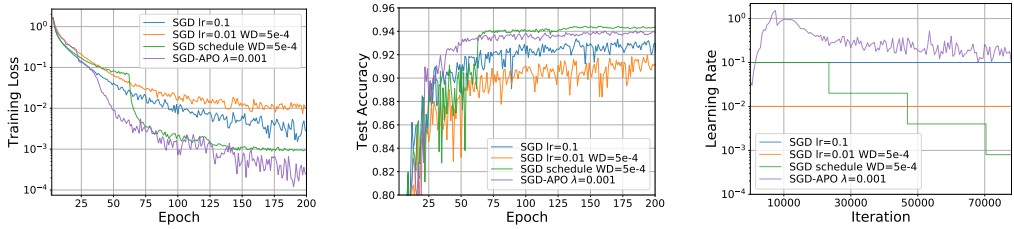

Figure 6: **SGD with weight decay compared to SGD-APO without weight decay, on CIFAR-10.**

### 5.3 BATCH NORMALIZATION AND WEIGHT DECAY

Batch normalization (BN) (Ioffe & Szegedy, 2015) is a widely used technique to speed up neural net training. Networks with BN are commonly trained with weight decay. It was shown that the effectiveness of weight decay for networks with BN is not due to the regularization, but due to the fact that weight decay affects the scale of the network weights, which changes the effective learning rate (Zhang et al., 2018; Hoffer et al., 2018; van Laarhoven, 2017). In particular, weight decay decreases the scale of the weights, which increases the effective learning rate; if one uses BN without regularizing the norm of the weights, then the weights can grow without bound, pushing the effective learning rate to 0. Here, we show that using APO to tune learning rates allows for effective training of BN networks without using weight decay. In particular, we compared SGD-APO without weight decay and SGD with weight decay 5e-4. Figure 6 shows that SGD-APO behaved better than SGD with a fixed learning rate, and achieved comparable performance as SGD with a manual schedule.

### 6 CONCLUSIONS

We introduced amortized proximal optimization (APO), a method for online adaptation of optimization hyperparameters, including global and per-layer learning rates, and damping parameters for approximate second-order methods. We evaluated our approach on real-world neural network optimization tasks—training MLP and CNN models—and showed that it converges faster and generalizes better than optimal fixed learning rates. Empirically, we showed that our method overcomes short horizon bias and performs well with sensible default values for the meta-optimization parameters.

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

## A    PROOF OF THEOREM 1

We first introduce the following lemma:

**Lemma 1.** *Assume the manifold is smooth with $\mathcal{C}$-bounded curvature, the gradient norm of loss function $\mathcal{L}$ is upper bounded by $\mathcal{G}$. If the effective gradient at point $\mathbf{Z}_k \in \mathcal{M}$ is $g_k$, then for any $0 < \gamma \leq \frac{1}{4\mathcal{C}\mathcal{G}}$ there exist $\mathbf{Z} \in \mathcal{M}$ such that*

$$\langle \nabla_{\mathbf{Z}} \mathcal{L}(\mathbf{Z}_k), \mathbf{Z} - \mathbf{Z}_k \rangle + \frac{1}{2\gamma} \|\mathbf{Z} - \mathbf{Z}_k\|_F^2 \leq -\frac{\gamma}{4} \|g_k\|^2.$$

*Proof.* We construct the $\mathbf{Z}$ satisfying the above inequality. Consider the following point in $\mathbb{R}^d$:

$$\mathbf{Y} \triangleq \mathbf{Z}_k - \gamma g_k.$$

Also, consider a geodesic

$$v : [0, \gamma \|g_k\|] \to \mathcal{M}$$

such that

$$\|\dot{v}\| = 1,$$

$$\dot{v}(0) = \frac{g_k}{\|g_k\|}.$$

Define point

$$\mathbf{Z} \triangleq v(\gamma \|g_k\|) \in \mathcal{M}.$$

We show that $\mathbf{Z}$ is a point satisfying the inequality in the lemma. Firstly, we notice that

$$\|\mathbf{Y} - \mathbf{Z}\| \leq \frac{\mathcal{C}}{2}\gamma^2 \|g_t\|^2.$$

This is because when we introduce the extra curve $\tilde{v} : [0, \gamma \|g_k\|] \to \mathbb{R}^d$ going directly from $\mathbf{Z}_k$ to $\mathbf{Y}$ with unit speed, there is

$$
\begin{aligned}
\|\mathbf{Y} - \mathbf{Z}\| &= \|v(\gamma \|g_t\|) - \tilde{v}(\gamma \|g_t\|)\| \\
&= \left\| \int_{t=0}^{\gamma\|g_t\|} \dot{v}(t)dt - \int_{t=0}^{\gamma\|g_t\|} \dot{\tilde{v}}(t) \right\| dt \\
&\leq \int_{t=0}^{\gamma\|g_t\|} \left\| \dot{v}(t) - \dot{\tilde{v}}(t) \right\| dt \\
&= \int_{t=0}^{\gamma\|g_t\|} \left\| \int_{t'=0}^{t} \ddot{v}(t')dt' - \int_{t'=0}^{t} \ddot{\tilde{v}}(t')dt' \right\| dt \\
&= \int_{t=0}^{\gamma\|g_t\|} \left\| \int_{t'=0}^{t} \ddot{v}(t')dt' \right\| dt \\
&\leq \int_{t=0}^{\gamma\|g_t\|} \int_{t'=0}^{t} \|\ddot{v}(t')\| \, dt'dt \\
&\leq \int_{t=0}^{\gamma\|g_t\|} \int_{t'=0}^{t} \mathcal{C} \, dt'dt \\
&= \frac{\mathcal{C}}{2}\gamma^2 \|g_t\|^2.
\end{aligned}
$$

Here we use the fact that $\ddot{\tilde{v}} = 0$ and $\|\ddot{v}\| \leq \mathcal{C}$. Therefore we have

$$
\begin{aligned}
& \langle \nabla_{\mathbf{Z}} \mathcal{L}(\mathbf{Z}_k), \mathbf{Z} - \mathbf{Z}_k \rangle + \frac{1}{2\gamma} \|\mathbf{Z} - \mathbf{Z}_k\|_F^2 \\
={} & \langle \nabla_{\mathbf{Z}} \mathcal{L}(\mathbf{Z}_k), (\mathbf{Z} - \mathbf{Y}) + (\mathbf{Y} - \mathbf{Z}_k) \rangle + \frac{1}{2\gamma} \|(\mathbf{Z} - \mathbf{Y}) + (\mathbf{Y} - \mathbf{Z}_k)\|_F^2 \\
\leq{} & \langle \nabla_{\mathbf{Z}} \mathcal{L}(\mathbf{Z}_k), \mathbf{Z} - \mathbf{Y} \rangle + \langle \nabla_{\mathbf{Z}} \mathcal{L}(\mathbf{Z}_k), \mathbf{Y} - \mathbf{Z}_k \rangle + \frac{1}{2\gamma} \|\mathbf{Z} - \mathbf{Y}\|_F^2 + \frac{1}{2\gamma} \|\mathbf{Y} - \mathbf{Z}_k\|_F^2 \\
={} & -\frac{\gamma}{2} \|g_k\|^2 + \langle \nabla_{\mathbf{Z}} \mathcal{L}(\mathbf{Z}_k), \mathbf{Z} - \mathbf{Y} \rangle + \frac{1}{2\gamma} \|\mathbf{Z} - \mathbf{Y}\|_F^2 \\
\leq{} & -\frac{\gamma}{2} \|g_k\|^2 + \frac{\mathcal{CG}}{2} \gamma^2 \|g_k\|^2 + \frac{\mathcal{C}^2}{8} \gamma^3 \|g_k\|^4 \\
\leq{} & -\frac{\gamma}{2} \|g_k\|^2 + \frac{\mathcal{CG}}{2} \gamma^2 \|g_k\|^2 + \frac{\mathcal{C}^2 \mathcal{G}^2}{8} \gamma^3 \|g_k\|^2 .
\end{aligned}
$$

Here the first equality is by introducing the extra $Y$, the first inequality is by triangle inequality, the second equality is by the definition of $g_k$ being $\nabla_{\mathbf{Z}} \mathcal{L}(\mathbf{Z}_k)$ projecting onto a plane, the second inequality is due to the above bound of $\|\mathbf{Y} - \mathbf{Z}\|$, the last inequality is due to $\|g_k\| \leq \|\nabla_{\mathbf{Z}} \mathcal{L}(\mathbf{Z}_k)\|$.

Since $\gamma \leq \frac{1}{4\mathcal{CG}}$, there is therefore

$$
\langle \nabla_{\mathbf{Z}} \mathcal{L}(\mathbf{Z}_k), \mathbf{Z} - \mathbf{Z}_k \rangle + \frac{1}{2\gamma} \|\mathbf{Z} - \mathbf{Z}_k\|_F^2 \leq -\frac{\gamma}{4} \|g_k\|^2 ,
$$

which completes the proof. $\qquad\square$

Now we return to the proof of Theorem 1:

*Proof.* For the ease of notation, we denote the effective gradient at iteration $k$ as $g_k$. For one iteration, there is

$$
\begin{aligned}
\mathcal{L}(\mathbf{Z}_{k+1}) &\leq \mathcal{L}(\mathbf{Z}_k) + \langle \nabla_{\mathbf{Z}} \mathcal{L}(\mathbf{Z}_k), \mathbf{Z}_{k+1} - \mathbf{Z}_k \rangle + \frac{L}{2} \|\mathbf{Z}_{k+1} - \mathbf{Z}_k\|^2 \\
&\leq \mathcal{L}(\mathbf{Z}_k) + \langle \nabla_{\mathbf{Z}} \mathcal{L}(\mathbf{Z}_k), \mathbf{Z}_{k+1} - \mathbf{Z}_k \rangle + 2\lambda \|\mathbf{Z}_{k+1} - \mathbf{Z}_k\|^2 \\
&\leq \mathcal{L}(\mathbf{Z}_k) - \frac{1}{16\lambda} \|g_k\|^2 .
\end{aligned}
$$

Here the first inequality is due to the Lipschitz continuity and the fact that total loss equals to the sum of all loss functions, and the second inequality is due to $\lambda \geq \frac{L}{4}$, the third inequality is due to Lemma 1 with $\gamma = \frac{1}{4\lambda}$. (When $\lambda \geq \mathcal{CG}$, there is naturally $\gamma \leq \frac{1}{4\mathcal{CG}}$, satisfying the assumptions in Lemma 1.)

So we have

$$
\mathcal{L}(\mathbf{Z}_{k+1}) \leq \mathcal{L}(\mathbf{Z}_k) - \frac{1}{16\lambda} \|g_k\|^2 .
$$

Telescoping, there is

$$
\|g_t^*\|^2 \leq \frac{1}{T} \sum_{t=0}^{T-1} \|g_t\|^2 \leq \frac{16\lambda}{T} \left[ \mathcal{L}(\mathbf{Z}_0) - \mathcal{L}^* \right] .
$$

$\qquad\square$

## B  PROOF OF THEOREM 2

*Proof.* For notational convenience, we think of $\mathbf{Z}$ as a vector rather than a matrix in this proof.

The Hessian $\nabla^2 \mathcal{L}(\mathbf{Z})$ is therefore a block diagonal matrix, where each block is the Hessian of loss on a single data.

First, we notice the following equation:

$$
\begin{aligned}
&\underset{\mathbf{Z} \in \mathcal{M}}{\arg\min} \left[ \langle \nabla_{\mathbf{Z}} \mathcal{L}(\mathbf{Z}_k), \mathbf{Z} - \mathbf{Z}_k \rangle + \frac{1}{2} \left\| \mathbf{Z} - \mathbf{Z}_k \right\|_H^2 \right] \\
&= \underset{\mathbf{Z} \in \mathcal{M}}{\arg\min} \left[ \langle \nabla_{\mathbf{Z}} \mathcal{L}(\mathbf{Z}_k), \mathbf{Z} - \mathbf{Z}_k \rangle + \frac{1}{2} (\mathbf{Z} - \mathbf{Z}_k)^T \nabla^2 \mathcal{L}(\mathbf{Z}_k)(\mathbf{Z} - \mathbf{Z}_k) \right] \\
&= \underset{\mathbf{Z} \in \mathcal{M}}{\arg\min} \frac{1}{2} \left[ \left\| \mathbf{Z} - \left( \mathbf{Z}_k - \nabla^2 \mathcal{L}(\mathbf{Z}_k)^{-1} \nabla \mathcal{L}(\mathbf{Z}_k) \right) \right\|_{\nabla^2 \mathcal{L}(\mathbf{Z}_k)}^2 - \left\| \nabla \mathcal{L}(\mathbf{Z}_k) \right\|_{[\nabla^2 \mathcal{L}(\mathbf{Z}_k)]^{-1}}^2 \right] \\
&= \underset{\mathbf{Z} \in \mathcal{M}}{\arg\min} \left\| \mathbf{Z} - \left( \mathbf{Z}_k - \nabla^2 \mathcal{L}(\mathbf{Z}_k)^{-1} \nabla \mathcal{L}(\mathbf{Z}_k) \right) \right\|_{\nabla^2 \mathcal{L}(\mathbf{Z}_k)}^2 .
\end{aligned}
$$

Here

$$
\|v\|_A^2 \triangleq v^T A v
$$

is the norm of vector $v$ defined by the positive definite matrix $A$. $\left[ \nabla^2 \mathcal{L}(\mathbf{Z}_k) \right]^{-1}$ is the inverse of positive definite matrix $\nabla^2 \mathcal{L}(\mathbf{Z}_k)$, therefore also positive definite.

As a result of the above equivalence, one step of Proximal Newton Method can be written as:

$$
\mathbf{Z}_{k+1} = \underset{\mathbf{Z} \in \mathcal{M}}{\arg\min} \left\| \mathbf{Z} - \left( \mathbf{Z}_k - \nabla^2 \mathcal{L}(\mathbf{Z}_k)^{-1} \nabla \mathcal{L}(\mathbf{Z}_k) \right) \right\|_{\nabla^2 \mathcal{L}(\mathbf{Z}_k)}^2 .
$$

Since $\mathbf{Z}^* \in \mathcal{M}$ by assumption, there is:

$$
\left\| \mathbf{Z}_{k+1} - \left( \mathbf{Z}_k - \nabla^2 \mathcal{L}(\mathbf{Z}_k)^{-1} \nabla \mathcal{L}(\mathbf{Z}_k) \right) \right\|_{\nabla^2 \mathcal{L}(\mathbf{Z}_k)} \leq \left\| \mathbf{Z}^* - \left( \mathbf{Z}_k - \nabla^2 \mathcal{L}(\mathbf{Z}_k)^{-1} \nabla \mathcal{L}(\mathbf{Z}_k) \right) \right\|_{\nabla^2 \mathcal{L}(\mathbf{Z}_k)} .
$$

Now we have the following inequality for one iteration:

$$
\begin{aligned}
&\left\| \mathbf{Z}_{k+1} - \mathbf{Z}^* \right\|_{\nabla^2 \mathcal{L}(\mathbf{Z}_k)} \\
&\leq \left\| \mathbf{Z}_{k+1} - \left( \mathbf{Z}_k - \nabla^2 \mathcal{L}(\mathbf{Z}_k)^{-1} \nabla \mathcal{L}(\mathbf{Z}_k) \right) \right\|_{\nabla^2 \mathcal{L}(\mathbf{Z}_k)} + \left\| \mathbf{Z}^* - \left( \mathbf{Z}_k - \nabla^2 \mathcal{L}(\mathbf{Z}_k)^{-1} \nabla \mathcal{L}(\mathbf{Z}_k) \right) \right\|_{\nabla^2 \mathcal{L}(\mathbf{Z}_k)} \\
&\leq 2 \left\| \mathbf{Z}^* - \left( \mathbf{Z}_k - \nabla^2 \mathcal{L}(\mathbf{Z}_k)^{-1} \nabla \mathcal{L}(\mathbf{Z}_k) \right) \right\|_{\nabla^2 \mathcal{L}(\mathbf{Z}_k)} \\
&= 2 \left\| \mathbf{Z}^* - \mathbf{Z}_k + \nabla^2 \mathcal{L}(\mathbf{Z}_k)^{-1} (\nabla \mathcal{L}(\mathbf{Z}_k) - \nabla \mathcal{L}(\mathbf{Z}^*) \right\|_{\nabla^2 \mathcal{L}(\mathbf{Z}_k)} \\
&\leq \frac{2}{\sqrt{\mu}} \left\| \nabla^2 \mathcal{L}(\mathbf{Z}_k)(\mathbf{Z}_k - \mathbf{Z}^*) - \nabla \mathcal{L}(\mathbf{Z}_k) + \nabla \mathcal{L}(\mathbf{Z}^*) \right\| .
\end{aligned}
$$

Here the first inequality is because of triangle inequality, the second inequality is due to the previous result, the equality is because $\nabla \mathcal{L}(\mathbf{Z}^*) = 0$, the last inequality is because of the strong convexity.

By the Lipschitz continuity of the Hessian, we have:

$$
\begin{aligned}
&\left\| \nabla^2 \mathcal{L}(\mathbf{Z}_k)(\mathbf{Z}_k - \mathbf{Z}^*) - \nabla \mathcal{L}(\mathbf{Z}_k) + \nabla \mathcal{L}(\mathbf{Z}^*) \right\| \\
&\leq \sum_{i=1}^{N} \left\| \nabla^2 \ell(\mathbf{z}_{k,i})(\mathbf{z}_{k,i} - \mathbf{z}_i^*) - \nabla \mathcal{L}(\mathbf{z}_{k,i}) + \nabla \mathcal{L}(\mathbf{z}_i^*) \right\| \\
&\leq \frac{L_H}{2} \sum_{i=1}^{N} \left\| \mathbf{z}_{k,i} - \mathbf{z}_i^* \right\|^2 \\
&= \frac{L_H}{2} \left\| \mathbf{Z}_k - \mathbf{Z}^* \right\|^2 .
\end{aligned}
$$

Therefore, we have:

$$\|\mathbf{Z}_{k+1} - \mathbf{Z}^*\| \le \frac{1}{\sqrt{\mu}} \|\mathbf{Z}_{k+1} - \mathbf{Z}^*\|_{\nabla^2 \mathcal{L}(\mathbf{Z}_k)} \le \frac{L_H}{\mu} \|\mathbf{Z}_k - \mathbf{Z}^*\|^2.$$

□

## C  MULTIPLE OPTIMIZATION HYPERPARAMETERS & PER-LAYER TUNING

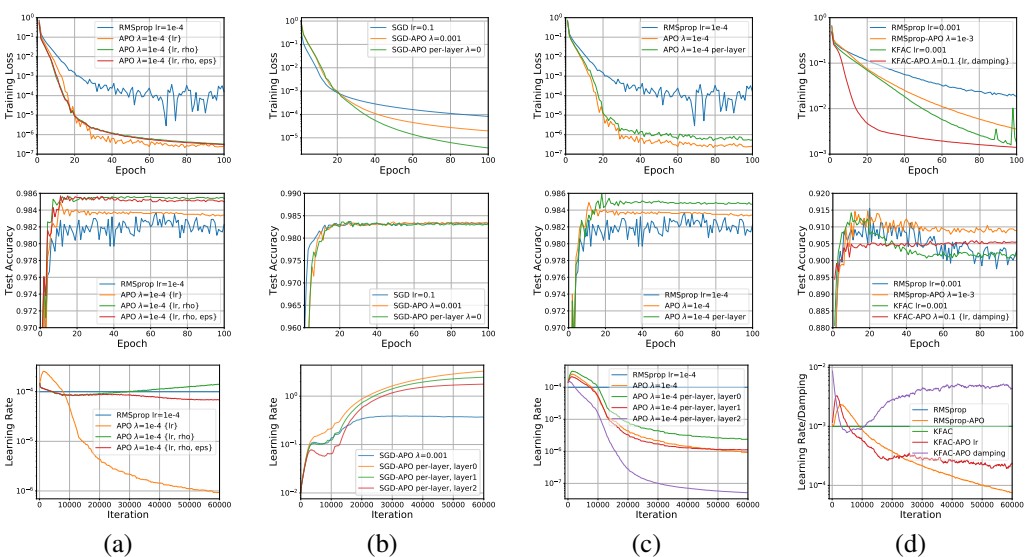

Figure 7: (a) Tuning multiple RMSprop parameters from $\{\eta, \rho, \epsilon\}$ on MNIST. (b) Tuning per-layer learning rates for SGD on MNIST. (c) Tuning per-layer learning rates for RMSprop on MNIST. (d) K-FAC and RMSprop on FashionMNIST.

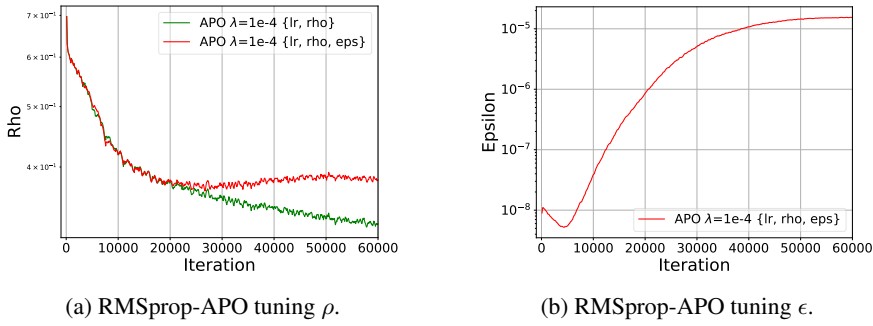

(a) RMSprop-APO tuning $\rho$.  (b) RMSprop-APO tuning $\epsilon$.

Figure 8: Adaptation of $\rho$ and $\epsilon$ using RMSprop-APO on MNIST.

Here we highlight the ability of APO to tune several optimization hyperparameters simultaneously. We used APO to adapt all of the RMSprop hyperparameters $\{\eta, \rho, \epsilon\}$. As shown in Figure 7(a), tuning $\rho$ and $\epsilon$ in addition to the learning rate $\eta$ can stabilize training. We also used APO to adapt per-layer learning rates. Figure 7(b) shows the per-layer learning rates tuned by APO, when using SGD on MNIST. Figure 7(c) uses APO to tune per-layer learning rate of RMSprop on MNIST. Figure 8 shows the adaptation of the additional $\rho$ and $\epsilon$ hyperparameters of RMSprop, for training an MLP on MNIST. Tuning per-layer learning rates is a difficult optimization problem, and we found that it was useful to use a smaller meta learning rate of 0.001 and perform meta-updates more frequently.

**K-FAC.** We also used APO to train a convolutional network on the FashionMNIST dataset (Xiao et al., 2017). The network we use consists of two convolutional layers with 16 and 32 filters respectively, both with kernel size 5, followed by a fully-connected layer. The results are shown in Figure 7(d), where we also compare K-FAC to hand-tuned RMSprop and RMSprop-APO on the same problem. We find that K-FAC with a fixed learning rate outperforms RMSprop-APO, while K-FAC-APO substantially outperforms K-FAC. The results are shown in Figure 7(d). We also show the adaptation of both the learning rate and damping coefficient for K-FAC-APO in Figure 7(d).

## D    ADDITIONAL EXPERIMENTS ON ROSENBROCK

In this section, we present additional experiments on the Rosenbrock problem. We show in Figure 9 that APO converges quickly from different starting points on the Rosenbrock surface.

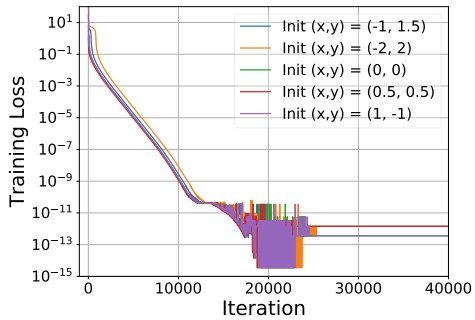

(a) RMSprop-APO training convergence from different initial (x,y) positions on the Rosenbrock surface.

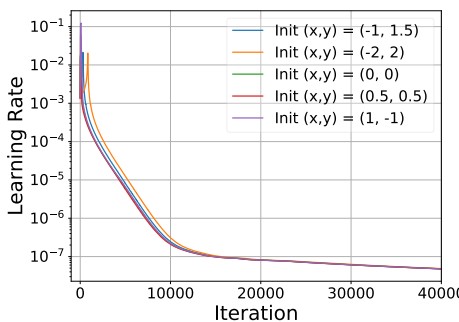

(b) RMSprop-APO learning rate adaptation from different initial (x,y) positions on the Rosenbrock surface.

Figure 9: RMSprop-APO convergence from different initializations on the Rosenbrock surface. We find that RMSprop-APO achieves very low training loss starting from any of the points $(x, y) \in \{(-1, 1.5), (-2, 2), (0, 0), (0.5, 0.5), (1, -1)\}$.

## E    ROBUSTNESS TO INITIAL LEARNING RATE

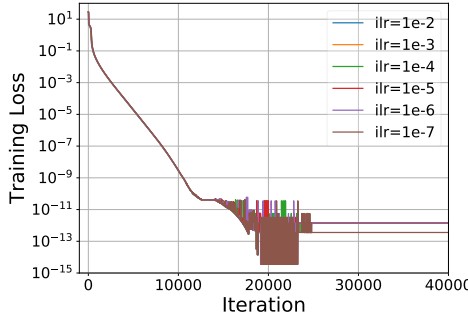

(a) RMSprop-APO training convergence with different initial learning rates.

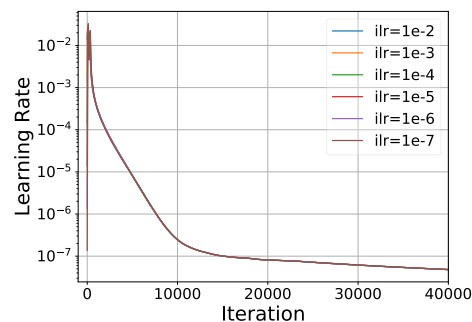

(b) RMSprop-APO learning rate adaptation with different initial learning rates.

Figure 10: RMSprop-APO performance on Rosenbrock, starting with different initial learning rates for the base optimizer.

In this section we show that APO is robust to the choice of initial learning rate of the base optimizer. With a suitable meta learning rate, APO quickly adapts many different initial learning rates to the same range, after which the learning rate adaptation follows a similar trajectory. Thus, APO helps to alleviate the difficulty involved in selecting an initial learning rate.

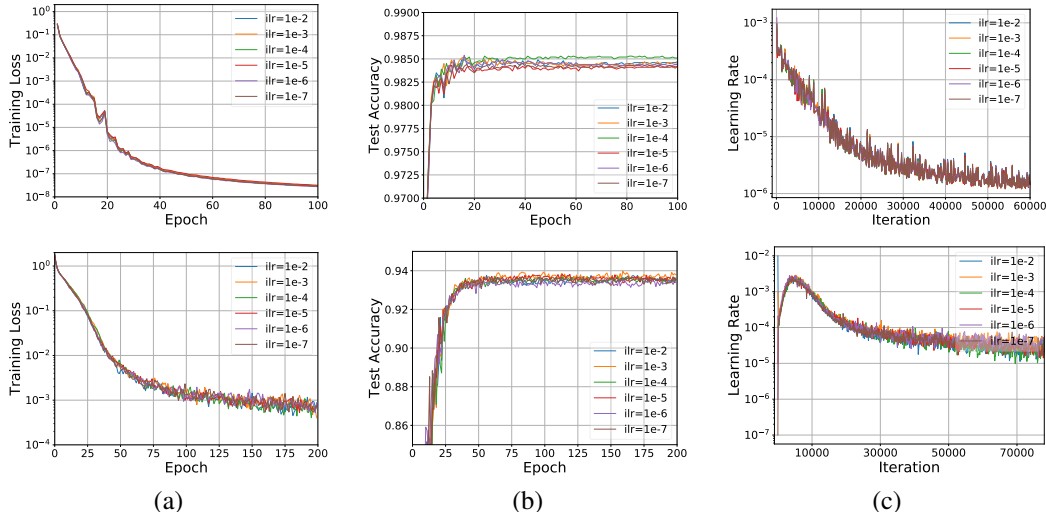

Figure 11: **Robustness to initial learning rates on MNIST and CIFAR-10. Top row:** RMSprop-APO tuning an MLP on MNIST. **Bottom row:** RMSprop-APO tuning ResNet34 on CIFAR-10. (a) Training loss; (b) test accuracy; (c) learning rate adaptation.

First, we used RMSprop-APO to optimize Rosenbrock, starting with a wide range of initial learning rates; we see in Figure 10 that the training losses and learning rates are nearly identical between all these experiments.

Next, we trained an MLP on MNIST and ResNet34 on CIFAR-10 using RMSprop-APO, with the learning rate of the base optimizer initialized to 1e-2, 1e-3, 1e-4, 1e-5, 1e-6, and 1e-7. We used the default meta learning rate 0.1. As shown in Figure 11, the training loss, test accuracy, and learning rate adaptation are nearly identical using these initial learning rates, which span 5 orders of magnitude.

## F  THE NOISY QUADRATIC PROBLEM

In this section we apply APO to the noisy quadratic problem investigated in (Wu et al., 2018; Schaul et al., 2013), and demonstrate that APO overcomes the short horizon bias problem. We optimize a quadratic function

$$f(x) = x^T H x,$$

where $x \in \mathbb{R}^{1000}$, $H$ is a diagonal matrix $H = diag\{h_1, h_2, \cdots, h_{1000}\}$, with eigenvalues $h_i$ evenly distributed in interval $[0.01, 1]$. Initially, we set $x$ with each dimension being 100. For each iteration, we can access the noisy version of the function, i.e., the gradient and function value of function

$$\tilde{f}(x) = (x - c)^T H (x - c).$$

Here $c$ is the vector of noise: each dimension of $c$ is independently randomly sampled from a normal distribution at each iteration, and the variance of dimension $i$ is set to be $\frac{1}{h_i}$. For SGD, we consider the following four learning rate schedules: optimal schedule, exponential schedule, linear schedule and a fixed learning rate. For SGD with APO, we directly use function $\tilde{f}$ as the loss approximation $h$, use Euclidean distance norm square as the dissimilarity term $\mathcal{D}$, and consider the following schedules for $\lambda$: optimal schedule(with $\lambda \geq 0$), exponential schedule, linear schedule and a fixed $\lambda$. We calculate the optimal parameter for each schedule of both algorithms so as to achieve a minimal function value at the end of 300 iterations. We optimize the schedules with 10000 steps of Adam and learning rate 0.001 after unrolling the entire 300 iterations.

The function values at the end of 300 iterations with each schedule are shown in Table 2.

Figure 12 plots the training loss and learning rate of SGD during the 300 iterations under optimal schedule, figure 13 plots the training loss and $\lambda$ under optimal schedule for SGD with APO. It can be

| | Optimal | Exponential | Linear | Fix |
|---|---|---|---|---|
| **SGD** | 3.3 | 4.6 | 12.5 | 81.9 |
| **SGD + APO** | 4.69 | 6.73 | 15.6 | 80.8 |

Table 2: Loss at the end of 300 iterations with optimized schedules.

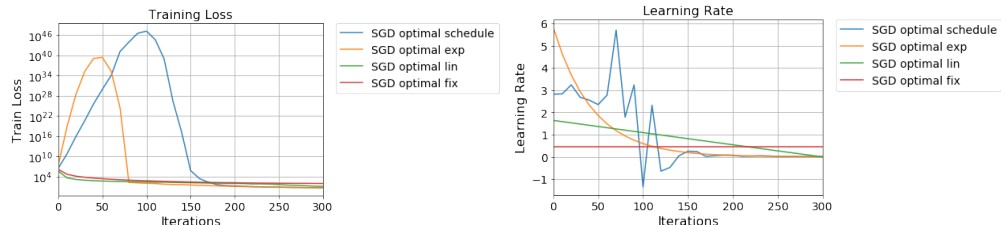

(a) Loss of different optimal schedules.  (b) Learning rate of different optimal schedules.

Figure 12: Loss and learning rate for SGD.

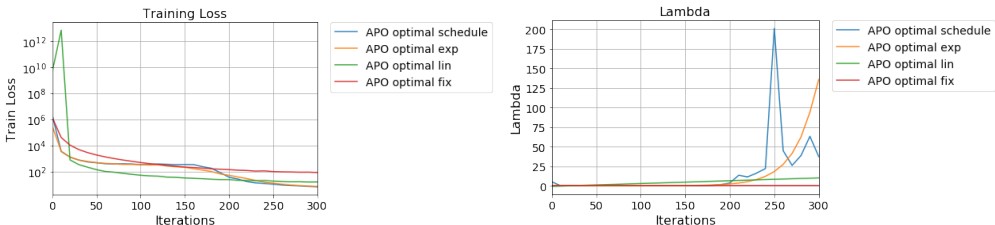

(a) Loss of different optimal schedules.  (b) Learning rate of different optimal schedules.

Figure 13: Loss and learning rate for APO.

seen that SGD with APO achieves almost the same training loss as optimal SGD for noisy quadratics task. This indicates that APO doesn't suffer from the short-horizon bias mentioned in (Wu et al., 2018).

## G    ADAM EXPERIMENTS

Adam (Kingma & Ba, 2015) is an adaptive optimization algorithm widely used for training neural networks, which can be seen as RMSProp with momentum. The update rule is given by:

$$m_t \leftarrow \beta_1 m_{t-1} + (1 - \beta_1) g_t \tag{22}$$

$$v_t \leftarrow \beta_2 v_{t-1} + (1 - \beta_2) g_t^2 \tag{23}$$

$$\hat{m}_t \leftarrow m_t / (1 - \beta_1^t) \tag{24}$$

$$\hat{v}_t \leftarrow v_t / (1 - \beta_2^t) \tag{25}$$

$$\theta_t \leftarrow \theta_{t-1} - \eta \hat{m}_t / (\sqrt{\hat{v}_t} + \epsilon) \tag{26}$$

where $g_t = \nabla_\theta \ell_t(\theta_{t-1})$.

Similar to SGD with Nesterov momentum, we fixed the $\beta_1$ and $\beta_2$ for Adam, and used APO to tune the global learning rate $\eta$. We tested Adam-APO with a ResNet34 network on CIFAR-10 dataset, and compared it with both Adam with fixed learning rate and Adam with a learning rate schedule where the learning rate is initialized to 1e-3 and is decayed by a factor of 5 every 60 epochs. Similarly to SGD with momentum, we found that Adam generally benefits from larger values of $\lambda$. Thus, we recommend performing a grid search over $\lambda$ values from 1e-2 to 1. As shown in Figure 14, APO improved both the training loss and test accuracy compared to the fixed learning rate, and achieved comparable performance as the manual learning rate schedule.

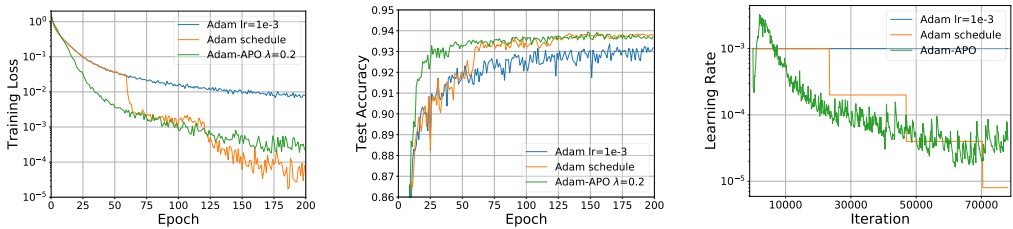

Figure 14: **Adam results on CIFAR-10.** We compare Adam with a fixed learning rate and a manual learning rate schedule to Adam-APO.

## H  COMPARISON WITH POPULATION-BASED TRAINING

Population-based training (PBT) (Jaderberg et al., 2017) is an approach to hyperparameter optimization that trains a *population* of $N$ neural networks simultaneously: each network periodically evaluates its performance on a target measure (e.g., the training loss); poorly-performing networks can exploit better-performing members of the population by cloning the weights of the better network, copying and perturbing the hyperparameters used by the better network, and resuming training. In this way, a single model can essentially experience multiple hyperparameter settings during training; in particular, we are interested in evaluating the learning rate schedule found using PBT.

Here, we used PBT to tune the learning rate for RMSprop, to optimize a ResNet34 model on CIFAR-10. For PBT, we used a population of size 4 (which we found to perform better than a population of size 10), and used a perturbation strategy that consists of randomly multiplying the learning rate by either 0.8 or 1.2. In PBT, one can specify the probability with which to re-sample a hyperparameter value from an underlying distribution. We found that it was critical to set this to 0; otherwise, the learning rate could jump from small to large values and cause instability in training. Figure 15 compares PBT with APO; we show the best training loss achieved by any of the models in the PBT population, as a function of wall-clock time. For a fair comparison between these methods, we ran both PBT and APO using 1 GPU. We see that APO outperforms PBT, achieving a training loss an order of magnitude smaller than PBT, and achieves the same test accuracy, much more quickly.

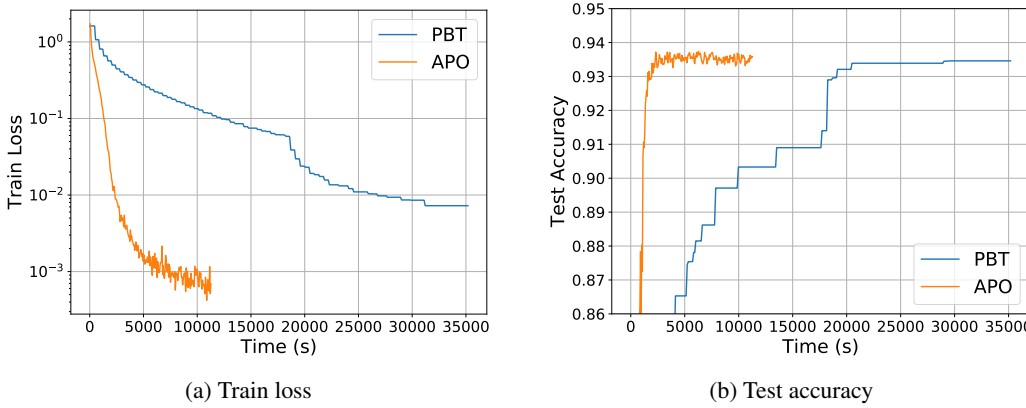

(a) Train loss                    (b) Test accuracy

Figure 15: Training loss and test accuracy of APO and PBT, as a function of wall-clock time.

