# OpenReview forum: "Online Hyperparameter Adaptation via Amortized Proximal Optimization"
_ICLR.cc/2019/Conference_

### Official Review · AnonReviewer3 · 2018-10-25
**Simple and intuitive idea but needs more clarification**

**Rating:** 7
**Confidence:** 3

**Review:**

I raised my rating. After the rebuttal.

- the authors address most of my concerns.
- it's better to show time v.s. testing accuracy as well. the per-epoch time for each method is different.
- anyway, the theory part acts still more like a decoration. as the author mentioned, the assumption is not realistic.

-------------------------------------------------------------
This paper presents a method to update hyper-parameters (e.g. learning rate) before updating of model parameters. The idea is simple but intuitive. I am conservative about my rating now, I will consider raising it after the rebuttal.

1. The focus of this paper is the hyper-parameter, please focus and explain more on the usage with hyper-parameters.
- no need to write so much in section 2.1, the surrogate is simple and common in optimization for parameters. After all, newton method and natural gradients method are not used in experiments.
- in section 2.2, please explain more how gradients w.r.t hyper-parameters are computed.

2. No need to write so much decorated bounds in section 3. The convergence analysis is on Z, not on parameters x and hyper-parameters theta. So, bounds here can not be used to explain empirical observations in Section 5.

3. Could authors explain the time complexity of inner loop in Algorithm 1? Does it take more time than that of updating model parameters?

4. Authors have done a good comparison in the context of deep nets.  However,
- could the authors compare with changing step-size? In most of experiments, the baseline methods, i.e. RMSProp are used with fixed rates. Is it better to decay learning rates for toy data sets? It is known that SGD with fixed step-size can not find the optimal for convex (perhaps, also simple) problems.
- how to tune lambda? it is an important hyper-parameter, but it is set without a good principle, e.g., "For SGD-APO, we used lambda = 0.001, while for SGDm-APO, we used lambda = 0.01", "while for RMSprop-APO, the best lambda was 0.0001". What are reasons for these?
- In Section 5.2, it is said lambda is tuned by grid-search. Tuning a good lambda v.s. tuning a good step-size, which one costs more?

---

> ### Author Response · Authors · 2018-11-26
> **Clarified writing and additional experiments**
>
> Thank you for your helpful comments. We have improved the writing to incorporate your feedback. We have also performed more experiments to compare APO to manual learning rate schedules.
>
> Q: Please explain more how gradients w.r.t hyper-parameters are computed.
>
> We implemented custom versions of the optimizers we consider (SGD, RMSprop, and K-FAC) that treat the optimization hyperparameters as variables in the computation graph for an optimization step. We then use automatic differentiation to compute the gradient of the meta-objective with respect to the hyperparameters (e.g., the learning rate).
>
>
> Q: Could authors explain the time complexity of inner loop in Algorithm 1? Does it take more time than that of updating model parameters?
>
> Each meta-optimization step requires approximately the same amount of computation as a parameter update for the model.
>
> By using the default meta learning rate suggested in our updated paper, we can amortize the meta-optimization by performing 1 meta-update for every K steps of the base optimization. We found that K=10 works well across our settings, while reducing the computational requirements of APO to just a small fraction more than the original training procedure.
>
> We have added a discussion of our meta-optimization setup and the efficiency of APO in Section 5 of the updated paper.
>
>
> Q: No need to write so much decorated bounds in section 3. The convergence analysis is on Z, not on parameters x and hyper-parameters theta. So, bounds here cannot be used to explain empirical observations in Section 5.
>
> The convergence of the network output Z directly indicates the rate of decrease of the loss function, which is exactly what we observe in practice. Although the assumption of a global optimization oracle is not realistic, we believe our theoretical justification provides insight into why the method works. One important takeaway from the theoretical analysis is that running gradient descent on output space can potentially accelerate the optimization (since the convergence bounds have better constants). This directly motivates the regularization term in our meta objective to be defined as the discrepancy of network outputs instead of the network parameters, which is essential to our technique.
>
>
> Q: Could the authors compare with changing step-size?
>
> Thank you for the suggestion. We have added comparisons with custom learning rate schedules for CIFAR-10 and CIFAR-100.
> We updated our results for CIFAR-10/100 using a larger network, ResNet34, instead of the VGG11 model used in the previous version, and we used a manual learning rate decay schedule where we trained for 200 epochs, decaying the learning rate by a factor of 5 three times during training.
>
> We found that APO is competitive with the custom schedule, achieving similar training loss and test accuracy. We provide results in our response to all reviewers at the top.
>
>
> Q: How to tune lambda? Tuning a good lambda v.s. tuning a good step-size, which one costs more?
>
> We tune lambda by performing a grid search over the range {1e-1, 1e-2, 1e-3, 1e-4, 1e-5}. Because each lambda value gives rise to a learning rate schedule, tuning lambda yields significantly more value than tuning a fixed learning rate. Instead of trying to come up with a custom learning rate schedule, which would require deciding how frequently to decay the learning rate, and by what factor it should be decayed, all one needs to do is perform a grid search over a fixed set of lambdas to find an automated schedule that is competitive with hand-designed schedules (which are the result of years of accumulated experience in the field).

---

### Official Review · AnonReviewer2 · 2018-11-04
**Update baselines**

**Rating:** 5
**Confidence:** 4

**Review:**

The paper proposes an approach to adapt hyperparameters online.
When learning rates are in focus, a convincing message would be to show that adaptation of learning rates is more efficient and simpler than their scheduling when tested on state-of-the-art architectures.
A. You demonstrate the results on CIFAR-10 for 10% error rates which corresponds to networks which are far from what is currently used in deep learning. Thus, it is hard to say whether the results are applicable in practice.
B. You don't schedule learning rates for your baseline methods except for a single experiment for some initial learning rate.
C. Your method involves a hyperparameter to be tuned which affects the shape of the schedule. This hyperparameter itself benefits from (requires?) some scheduling.

It would be interesting to see if the proposed method is competitive for training contemporary networks and w.r.t. simple schedule schemes. Online tuning of  hyperparameters is an important functionality and I hope your paper will make it more straightforward to use it in practice.


* Minor notes:

You mention that "APO converges quickly from different starting points on the Rosenbrock surface" but 10000 iterations is not quick at all for the 2-dimensional Rosenbrock, it is extremely slow compared to 100-200 function evaluations needed for Nelder-Mead to solve it. I guess you mean w.r.t. the original RMSprop.

---

> ### Author Response · Authors · 2018-11-26
> **Updated baselines, and comparison to learning rate schedules**
>
> Thank you for your helpful comments. We have addressed your concern about the baseline models and learning rate schedules in our updated paper.
>
> Q: It is hard to say whether the results are applicable in practice; need updates to baselines and comparison with learning rate schedules.
>
> We have updated the baselines in our paper for CIFAR-10 and CIFAR-100, using a larger, modern network, ResNet34, in place of the VGG11 model used previously. We also compared APO to manual learning rate decay schedules. For CIFAR-10/100, we trained for 200 epochs, decaying the learning rate by a factor of 5 three times during training. The ResNet34 with a custom learning rate decay schedule achieves 93-94% test accuracy on CIFAR-10 and ~74% test accuracy on CIFAR-100. We believe that this is a strong baseline, and shows the applicability of APO in practical settings.
>
> The final test accuracies of the updated model using SGD/SGDm with and without APO are:
>
>                                   | CIFAR-10 | CIFAR-100 |
> --------------------------+--------------+---------------+
> SGD (fixed lr)                92.97            72.69
> SGDm (fixed lr)            92.77            72.53
> SGD (decayed lr)          93.29            73.45
> SGDm (decayed lr)      93.53            73.80
> SGD-APO                       93.82            74.65
> SGDm-APO                   94.59            73.89
>
> The test accuracies using RMSprop and K-FAC with APO are shown in our response to all reviewers at the top.
> The results in these tables show that APO is competitive with manual schedules in terms of test accuracy. The updated figures in our paper show that APO is competitive with manual schedules both in terms of test accuracy and training loss.
>
>
> Q: Does the hyperparameter lambda itself benefit from some scheduling?
>
> In our updated paper we show that APO with a fixed lambda achieves comparable performance to manual learning rate decay schedules. While using a schedule for lambda can potentially further improve performance, a simple grid search over fixed lambda values already leads to strong performance, and has the advantage that it is easy to use in practice.
>
>
> Q: You mention that "APO converges quickly from different starting points on the Rosenbrock surface" but 10000 iterations is not quick at all for the 2-dimensional Rosenbrock, it is extremely slow compared to 100-200 function evaluations needed for Nelder-Mead to solve it. I guess you mean w.r.t. the original RMSprop.
>
> Yes, we intended to say that on Rosenbrock, RMSprop-APO converges quickly compared to baseline RMSprop; we have updated the paper to clarify this.

---

> ### Author Response · Authors · 2018-12-01
> **Updated baselines, and comparison to learning rate schedules (summary)**
>
> Thank you for your helpful feedback. We have incorporated your suggestions into the updated paper.
>
> Specifically, we have:
>
> * Updated the baseline model for CIFAR-10/100 from VGG11 to ResNet34.
> * Used manual learning rate decay schedules for the CIFAR-10/100 baselines. We obtained 93-94% test accuracy on CIFAR-10 (SGD/SGDm/RMSprop/K-FAC) and 73-74% test accuracy on CIFAR-100 (SGD/SGDm). All are compared to their APO variants, which performed as well or better. The final results are shown in the table in the response to all reviewers at the top.
> * Shown that APO is competitive with manual schedules both in terms of test accuracy and training loss with ResNet34. This demonstrates the practical applicability of APO for contemporary networks.
> * Updated Figure 2 on CIFAR-10 with SGD/SGDm/RMSprop, Figure 4 on CIFAR-100 with SGD/SGDm, and Figure 6 on CIFAR-10 with SGD. We also added Figure 3 on CIFAR-10 with K-FAC. Each figure compares the baseline optimizers with their APO variants.
>
> Thank you for having helped us improve the paper.

---

> > ### Comment · AnonReviewer2 · 2018-12-01
> > **Response**
> >
> > Thank you for your updates.
> >
> > 1) Please clarify why you didn't use weight decay for your CIFAR experiments except for the case without momentum? Weight decay is used in most experimental setups for CIFAR-10 and CIFAR-100. In other words, it seems that you used a weaker baseline which makes "showed that it converges faster and generalizes better than optimal fixed learning rates" too strong.
> >
> > 2) I don't think that ResNet34 is a state-of-the-art/"strong baseline" network. It is a tiny network. Wide ResNets showed better results (about 4% compared to about 6% for your ResNet)  back in 2016 (2.5 years ago). I was asking for better networks to make it easier to compare with known results.
> >
> > 3) Could you please be a bit more specific about "a small fraction" of the computational overhead and whether it is related to the size of the network. I guess that the size of the network might also affect some results, e.g., for layer-wise learning rates.
> >
> > 4) I think that the paper would benefit from a more detailed interpretation of the hyperparameter lambda. To which extent it is wrong/correct to view it as just a higher level learning rate (step-size, scaling factor) especially when only one hyperparameter, learning rate of the low-level optimizer is considered.
> >
> > 5) It would be nice to have an experiment similar to the one given in "No More Pesky Learning Rates" by Tom Schaul, Figure 2. This is definitely too late to ask it for this review but you may consider it for your future works.
> >
> > Thank you.

---

> > > ### Author Response · Authors · 2018-12-03
> > > **WideResNet-28-10, noisy quadratic problem, and lambda**
> > >
> > > Q: Weight decay
> > >
> > > We used weight decay 1e-5 for all SGD/SGDm/RMSprop experiments on CIFAR. We apologize that this was not clearly described in the paper, and we will add this information to the final version.
> > >
> > > Using a larger weight decay of 5e-4 improves the baselines for SGD and SGDm to 94.32% and 94.82%, respectively, on CIFAR-10. Our current results for SGD-APO (93.82%) and SGDm-APO (94.59%) are still comparable to their respective baselines. We also verified that SGD-APO and SGDm-APO perform well with weight decay 5e-4, achieving test accuracies 94.22% and 94.62%, respectively.
> > >
> > > In addition, we show in the WideResNet experiment below that APO performs well when using weight decay 5e-4.
> > >
> > >
> > > Q: Wide ResNets
> > >
> > > We didn't use Wide ResNets due to computational limitations. Our aim was to show that APO is competitive with good fixed learning rates and with manual learning rate schedules for networks with reasonable performance.
> > >
> > > We have obtained the following results with a 28-layer wide residual network with a widening factor of 10 (WideResNet-28-10). We followed the experimental setup from [1], training the model for 200 epochs with weight decay and the same learning rate schedule, that decays the learning rate by a factor of 5 three times during training. Our baseline result using SGD with momentum and a learning rate schedule achieves 96.01% test accuracy, and SGDm-APO achieves 95.48%. This shows that with this large network we achieve comparable results to a manual schedule.
> > >
> > > We used the standard weight decay 5e-4 (as in [1]), and we show the results in the table below. We found that APO outperforms fixed learning rates and achieves a test accuracy comparable to the manual schedule.
> > >
> > > CIFAR-10               | WideResNet-28-10 |
> > > ------------------------+---------------------------+
> > > SGD (fixed lr)                     89.42
> > > SGDm (fixed lr)                 92.60
> > > SGD (decayed lr)               94.73
> > > SGDm (decayed lr)           96.01
> > > SGD-APO                            94.84
> > > SGDm-APO                        95.48
> > >
> > > We will include these results in the final version.
> > >
> > >
> > > Q: Computational overhead
> > >
> > > Using the default meta-optimization scheme proposed in our paper, APO is ~1.3x as expensive as the baseline in practice. This cost could be reduced by performing less frequent meta-updates. In addition, our implementation is not optimized for performance, so this overhead could be reduced with careful engineering. In our current implementation, the overhead is somewhat affected by the size of the network and the speed of data loading (for the dissimilarity term).
> > >
> > >
> > > Q: Interpretation of lambda
> > >
> > > Lambda trades off the progress made on the current minibatch with the average change in the predictions on independently-sampled minibatches. A large value of lambda will emphasize that an update to the model parameters should not cause a large change in the model’s predictions on a separate minibatch, and thus will yield more conservative updates, while a small value of lambda would lead to more aggressive updates.
> > >
> > > When one uses APO to tune only the base learning rate, the hyperparameter lambda can be thought of as a parametrization of a learning rate schedule. Given a fixed base optimizer, a larger lambda corresponds to smaller learning rates in the schedule and vice versa. However, the exact mapping from lambda to a learning rate schedule differs for different base optimizers and network architectures. We found that this mapping is strongly related to the type of base optimizers: for SGD, SGDm, and RMSprop, the same lambda will lead to different learning rate schedules. We recommend using a grid search over lambda to find a schedule that performs well.
> > >
> > >
> > > Q: It would be nice to have an experiment similar to the one given in "No More Pesky Learning Rates" by Tom Schaul, Figure 2.
> > >
> > > We have performed this experiment, and will add the plot to the final version. We followed the experimental setup from [2], which analyzed a quadratic cost function based on [3]. In the two-dimensional experiment of the noisy quadratic problem, we observed that SGD-APO approaches the minimum in the low curvature direction faster than the myopic best learning rate, which suggests that APO does not suffer from short-horizon bias.
> > >
> > >
> > > [1] Sergey Zagoruyko and Nikos Komodakis. Wide Residual Networks. BMVC 2016.
> > > [2] Yuhuai Wu, Mengye Ren, Renjie Liao, Roger Grosse. Understanding Short-Horizon Bias in Stochastic Meta-Optimization. ICLR 2018.
> > > [3] Tom Schaul, Sixin Zhang, Yann LeCun. No More Pesky Learning Rates. ICML 2013.

---

### Official Review · AnonReviewer1 · 2018-11-05
**Interesting and Novel contribution - Some concerns that need to be answered regarding experiments and theory**

**Rating:** 6
**Confidence:** 4

**Review:**

Summary:
This paper introduces Amortized Proximal Optimization (APO) that optimizes a proximal objective at each optimization step. The optimization hyperparameters are optimized to best minimize the proximal objective.

The objective is represented using a regularization style parameter lambda and a distance metric D that, depending on its definition, reduces the optimization procedure to Gauss-Newton, General Gauss Newton or Natural Gradient Descent.

There are two key convergence results which are dependent on the meta-objective being optimized directly which, while not practical, gives some insight into the inner workings of the algorithm. The first result indicates strong convergence when using the Euclidean distance as the distance measure D. The second result shows strong convergence when D is set as the Bregman divergence.

The algorithm optimizes the base optimizer on a number of domains and shows state-of-the-art results over a grid search of the hyperparameters on the same optimizer.


Clarity and Quality: The paper is well written.

Originality: It appears to be a novel application of meta-learning. I wonder why the authors didn’t compare or mention optimizers such as ADAM and ADAGRAD which adapt their parameters on-the-fly as well. Also how does this compare to adaptive hyperparameter training techniques such as population based training?

Significance:
Overall it appears to be a novel and interesting contribution. I am concerned though why the authors didn’t compare to adaptive optimizers such as ADAM and ADAGRAD and how the performance compares with population based training techniques. Also, your convergence results appear to rely on strong convexity of the loss. How is this a reasonable assumption? These are my major concerns.

Question: In your experiments, you set the learning rate to be really low. What happens if you set it to be arbitrarily high? Can you algorithm recover good learning rates?

---

> ### Author Response · Authors · 2018-11-26
> **Addressed Adam, PBT, and initial learning rates**
>
> Thank you for your insightful comments. We have incorporated your suggestions into the revised version of the paper.
>
> Q: Relationship to optimizers with adaptive learning rates, and comparison between Adam and Adam-APO.
>
> While Adam and Adagrad are often described as having “adaptive learning rates,” they still have a global learning rate that is just as critical to tune as for SGD. In our experiments, we consider tuning the learning rate for RMSprop, which also maintains adaptive learning rates for each parameter, and is closely related to Adam/Adagrad. Adam is essentially RMSprop with momentum; APO can be applied to Adam by applying momentum on top of the updates computed by APO.
>
> To address your question about Adam, we added experiments for tuning the global learning rate of Adam with APO in appendix Section G, Figure 14, where Adam-APO achieves better performance than Adam with a fixed global learning rate, and achieves comparable performance as Adam with a manual schedule.
>
>
> Q: Comparison with population-based training (PBT)
>
> We have added a comparison between APO and PBT in appendix Section H, Figure 15.
>
> For population-based training, one must carefully select many hyperparameters, including the size of the population, the perturbation strategy (e.g., randomly perturb the learning rate by multiplying it by 1.2 or 0.8), the exploration interval (e.g., the number of training iterations to run before exploiting other members of the population). We used PBT and APO to tune the learning rate of RMSprop while training a ResNet34 model on CIFAR-10. For PBT, we used a population of size 4, and chose to exploit/explore after each epoch of training. We tried multiple exploration strategies, and found that it was critical to set the probability of resampling a learning rate from an underlying distribution to be 0; otherwise, the learning rates could jump from small to large values, and yield unstable training.
>
> In contrast, APO only requires a simple grid search over lambda, and all other hyperparameters can be kept at their default settings. We found that  APO substantially outperformed PBT, achieving a lower final training loss and equal test accuracy in much less wall-clock time; this shows the advantage of gradient-based methods for tuning learning rates, such as APO, compared to evolutionary methods based on random perturbations such as PBT.
>
>
> Q: The convergence results appear to rely on strong convexity of the loss. How is this a reasonable assumption?
>
> Note that we assume strong convexity of the loss as a function of the output units, not as a function of the weights. Hence, our assumption is fairly realistic in the neural net setting. The loss function on top of the network output is usually defined as a simple convex function; for instance, in regression, a common choice of loss function is the quadratic loss (i.e, the squared distance between the network output and the true label), which is strongly convex. In fact, even without assuming that the loss function is strongly convex and that the output manifold is dense, we are still able to show a fast convergence rate. In the updated version of the paper, we show that our algorithm with an oracle converges to stationary point globally with a fast rate, which provides insight into why APO works well.
>
>
> Q: In your experiments, you set the learning rate to be really low. What happens if you set it to be arbitrarily high? Can you algorithm recover good learning rates?
>
> APO is robust to the initial learning rate of the base optimizer, using the default meta learning rate suggested in our updated paper. We have added a section to the appendix in which we include RMSprop-APO experiments on Rosenbrock, MNIST, and CIFAR-10 to show that the training loss, test accuracy, and learning rate trajectories are nearly identical when starting with initial learning rates {1e-2, 1e-3, 1e-4, 1e-5, 1e-6, 1e-7}, spanning 5 orders of magnitude. Note that 1e-2 is quite a large initial learning rate for RMSprop.

---

### Public Comment · (anonymous) · 2018-11-10
**Concerns about experiments (more experiments need to be done!)**

Hi,

I have several questions about the experiments:

- How does APO compare to standard learning rate decay schedule (e.g., decay lr with a factor of 10 in the middle of training)?
- The reported numbers in terms of test performance on CIFAR-10 (~90%) and CIFAR-100 (~65%) are lower than my expectation. As I know, VGG16 can easily get >70% on CIFAR-100 with BN and data augmentation. Besides, I suggest the authors focusing on CIFAR-100, rather than CIFAR-10 and MNIST (too easy).
- Can you show more experiments for K-FAC since you mentioned K-FAC in abstract and introduction? The experiments of K-FAC on MNIST is far from convincing, you should at least show some experiments on CIFAR. Also, you mentioned in 5.3 that K-FAC-APO first decreases the damping, then gradually increases the damping later in the training. Does it really make sense? As argued by the original K-FAC paper, the damping would diminish later in the training since the quadratic approximation is accurate enough.

---

> ### Author Response · Authors · 2018-11-26
> **Thank you for your comment**
>
> Thank you for your helpful comments.
>
> Q: Comparison to manual learning rate decay schedule and improved baselines
>
> We have updated the baselines in our paper, using a ResNet34 network with manual learning rate schedule. This schedule achieves ~94% test accuracy on CIFAR-10 and ~74% test accuracy on CIFAR-100. We believe that this is a strong baseline, and show that APO achieves comparable and sometimes better performance as these manual schedules.
>
>
> Q: Additional K-FAC results
>
> We have added K-FAC results on CIFAR-10, in which we use APO to tune the learning rate and damping, and compare to K-FAC with a fixed learning rate as well as a manual decay schedule. We find that APO performs well when tuning the global learning rate, and that the training loss and test accuracy improve when we tune both the learning rate and damping coefficient.

---

### Author Response · Authors · 2018-11-26
**Updated paper with additional experiments and improved writing**

We thank all the reviewers for their insightful and helpful comments.

We made the following changes to the paper to address the reviewers’ concerns:

Updated Baselines and Comparison to Learning Rate Decay Schedules
-------------------------------------------------------------------------------------------------
We updated our results for CIFAR-10 and CIFAR-100 using a larger network, ResNet34, instead of the VGG11 model used in the previous version. We also compared APO to manual learning rate decay schedules. For CIFAR-10/100, we trained the ResNet34 for 200 epochs, decaying the learning rate by a factor of 5 three times during training.

The final test accuracies of the updated model with and without APO are:

                                  | CIFAR-10 | CIFAR-100 |
--------------------------+--------------+---------------+
SGD (fixed lr)                92.97            72.69
SGDm (fixed lr)            92.77            72.53
SGD (decayed lr)          93.29            73.45
SGDm (decayed lr)      93.53            73.80
SGD-APO                       93.82            74.65
SGDm-APO                   94.59            73.89

                                         | CIFAR-10 |
-------------------------------+--------------+
RMSprop (fixed lr)              92.00
RMSprop (decayed lr)        93.54
RMSprop-APO                     93.58

                                        | CIFAR-10 |
-------------------------------+--------------+
K-FAC (fixed lr)                     92.56
K-FAC (decayed lr)               94.25
K-FAC-APO {lr}                      93.91
K-FAC-APO {lr,damping}     94.51

Our manual learning rate decay schedules achieve test accuracies ~93-94% on CIFAR-10, which we believe are strong baselines.
In all cases, the learning rate schedules discovered by APO are competitive with the custom schedules.


Learning Rate Initialization
-------------------------------------
We added a section to the appendix (Section E) in which we show that APO is robust to the initial learning rate of the base optimizer. We perform experiments with RMSprop on Rosenbrock, MNIST, and CIFAR-10, and show that the training loss, test accuracy, and learning rate trajectories are nearly identical when using initial learning rates that range across 5 orders of magnitude.

Computational Efficiency
----------------------------------
Each meta-optimization step requires approximately the same amount of computation as a parameter update for the model.

By using a sufficiently large meta learning rate, we can amortize the meta-optimization by performing 1 meta-update for every K steps of the base optimization. We found that K=10 works well across our settings, while reducing the computational requirements of APO to just a small fraction more than the original training procedure.

Choosing Lambda
-------------------------
The only parameter that needs to be tuned in APO is lambda. The meta learning rate and meta update interval can be kept at our default values, which work well across many settings. Since each setting of lambda determines a learning rate schedule, tuning lambda is more valuable than tuning a fixed learning rate; it is equivalent to tuning a full learning rate schedule, for which the search space is much larger (i.e., to find a schedule manually, one must decide how often to decay the learning rate, and by what factor to decay each time).

---

> ### Author Response · Authors · 2018-11-26
> **Updated paper with additional experiments and improved writing (continued)**
>
> Comparison to Population-Based Training
> ---------------------------------------------------------
> We added Section H to the appendix comparing PBT with APO to adapt the learning rate for RMSprop while training a ResNet34 model on CIFAR-10.
>
> There are several important design decisions that must be made for PBT, including 1) the population size; 2) the exploration strategy; 3) the exploration frequency; and 4) the resampling probability. In particular, we found that it was critical to set the probability of resampling a learning rate value from an underlying hyperparameter distribution to 0; otherwise, the learning rate would jump from small to large values and cause training to become unstable.
>
> In contrast, APO only requires a simple grid search over lambda. We found that  APO substantially outperformed PBT, achieving a lower final training loss and equal test accuracy in much less wall-clock time. Because APO uses gradient-based optimization to tune the learning rate, it is more efficient than PBT, which is an evolutionary method that uses random perturbations to adapt the learning rate.
>
> Adam
> --------
> We added experiments for tuning the global learning rate of Adam with APO in appendix Section G, where Adam-APO achieves better performance than Adam with a fixed global learning rate, and is competitive with Adam with a manual schedule.
>
> K-FAC
> --------
> We added results for K-FAC on CIFAR-10. We compare K-FAC-APO to K-FAC with a fixed learning rate as well as a learning rate schedule. We find that APO performs well when tuning the global learning rate, and that the training loss and test accuracy improve when we tune both the learning rate and damping coefficient.

---

> > ### Author Response · Authors · 2018-11-27
> > **Updated paper with additional experiments and improved writing (continued)**
> >
> > In the current version of the paper, we have updated Figure 2 on CIFAR-10 with SGD/SGDm/RMSprop, Figure 4 on CIFAR-100 with SGD/SGDm, Figure 5 on SVHN with RMSprop, and Figure 6 on CIFAR-10 with SGD. We also added Figure 3 on CIFAR-10 with K-FAC. Each figure compares the baseline optimizers with their APO variants.
> >
> > We also added Sections E, G, and H to the Appendix, to show robustness to initial learning rates, Adam experiments, and a comparison to population-based training, respectively.

---

### Meta-Review · Area_Chair1 · 2018-12-16
**Interesting idea but does not make the bar.**

**Confidence:** 4
**Recommendation:** Reject

**Metareview:**

This paper proposes an amortized proximal optimization method to adapt optimization hyperparameters. Empirical results on many problems are performed.

Reviewers overall find the ideas interesting, however there still are some questions whether strong baselines are used in the experimental comparisons. The reviewers also point that the theoretical results are not useful ones since the assumptions are not satisfied in practice. One of the reviewer increased their score, but the other has maintained that the paper requires more work.

The presentation of the result is also a bit problematic; the font sizes in the figure are too small to read.

The paper contains interesting ideas, but it does not make the bar for acceptance in ICLR. Therefore I recommend a reject. I encourage the authors to resubmit this work after improving the presentation and experiments.